# Feedback cooling of fermionic atoms in optical lattices

Wenhua Zhao[1,2⋆], Ling-Na Wu[1,3†], Francesco Petiziol[1‡] and André Eckardt[1∘]

**1** Institut für Theoretische Physik, Technische Universität Berlin,
Hardenbergstraße 36, 10623 Berlin, Germany
**2** Max-Born-Institut, 12489 Berlin, Germany
**3** Center for Theoretical Physics & School of Physics and Optoelectronic Engineering,
Hainan University, Haikou, Hainan 570228, China

⋆ wenhua.zhao@physik.hu-berlin.de , † lingna.wu@hainanu.edu.cn ,
‡ f.petiziol@tu-berlin.de , ∘ eckardt@tu-berlin.de

## Abstract

We discuss the preparation of topological insulator states with fermionic ultracold atoms in optical lattices by means of measurement-based Markovian feedback control. The designed measurement and feedback operators induce an effective dissipative channel that stabilizes the desired insulator state, either in an exact way or approximately in the case where additional experimental constraints are assumed. Successful state preparation is demonstrated in one-dimensional insulators as well as for Haldane's Chern insulator, by calculating the fidelity between the target ground state and the steady state of the feedback-modified master equation. The fidelity is obtained numerically through exact diagonalization or via time evolution of the system with moderate sizes. For larger 2D systems, we compare the mean occupation of the single-particle eigenstates for the ground and steady state computed through mean-field kinetic equations.



# 1   Introduction

A topological insulator is a state where fermionic particles completely fill a topologically non-trivial energy band [1]. An important example is given by Chern insulators in two dimensions [2–4], corresponding to lattice versions of the integer quantum Hall effect [5, 6]. Here particles occupy bands characterized by a non-zero Chern number, giving rise to quantized response functions, such as the Hall conductivity or a circular dichroism with respect to driving-induced interband excitations.

In order to prepare a Chern insulator state in a quantum simulator of ultracold fermionic atoms in an optical lattice, two problems have to be addressed. On the one hand, topologically non-trivial Chern bands have to be engineered. This problem has been successfully addressed in a number of different experiments. Using Floquet engineering, both the paradigmatic square-lattice Harper-Hofstadter model as well as Haldane-type honeycomb lattices have been implemented [7–11]. On the other hand, a band-insulating state has to be prepared, where (at least) one topological band has to be filled completely with fermions. This second problem is not yet solved fully. Namely, since (unlike electronic systems in solid state) ultracold atoms are not coupled to a thermal bath, the Chern insulator state has to be prepared adiabatically, starting from the topologically-trivial regime prepared initially. This implies that a topological phase transition has to be crossed, where the energy gap of the band structure closes. The band touching necessarily leads to a deviation from the desired adiabatic passage, corresponding to interband excitations that subsequently remain in the system due to its isolation.

An appealing alternative to adiabatic passage is given by dissipative preparation methods [12–14], in which a dissipative process is identified and engineered whose unique steady-state is the target state. These ideas have been theoretically explored, for instance, for preparing nonequilibrium Bose-Einstein condensates [15, 16] and a range of topological states [17–25] including Floquet band insulators [22, 26] and fractional quantum Hall states [23, 24]. In experiments, engineered dissipation has been employed to stabilize entangled states of ions [27], Mott insulating states of bosons [28] and fermions [29] in optical lattice systems, and of photons in superconducting circuits [30]. This approach has the twofold advantage that, on the one hand, the target state would be prepared independently of the initial state and that the system would return to the target state whenever driven away from it.

In this work, we investigate the possibility to dissipatively prepare (topological) band insulators with ultracold atoms in optical lattices by means of Markovian feedback control [31–33]. Measurement-based feedback control has already been successfully implemented in various systems for tasks such as quantum state preparation and stabilization [34–57], cooling [58–65,65–68], simulating nonlinear dynamics [69–71], controlling dynamics [72–77] and manipulating phase transitions [78–82], among others. As a measurement-based approach, Markovian feedback control operates by continuously adding a signal-proportional feedback term to the system Hamiltonian [31–33]. This method has been used for the stabilization of arbitrary one-qubit quantum states [35, 48], the control of two-qubit entanglement [83–86], and the generation of optical and spin squeezing [82, 87]. In previous works by some of us, we have shown that Markovian feedback control can be used to cool bosonic atoms in a one-dimensional optical lattice [88], and for quantum engineering of a synthetic thermal bath [89] and heat-current-carrying states [90]. Here, we consider two-band fermionic models in one-(1D) and two-dimensional (2D) lattices at half filling. We first show that a mechanism able to dissipatively pump particles to the lower band is sufficient to prepare the desired state. Then, we discuss the experimental implementation of such an interband cooling process with measurement and homodyne-based feedback. We derive both an exact implementation and approximate ones that aim at favouring experimental feasibility. The approximate scheme further reveals interesting connections between the topological properties of the system, namely the unavailability of (exponentially) localized Wannier functions in topological bands, and the performance of the ground state preparation using only local measurements and feedback.

This paper is organized as follows. A brief introduction to Markovian feedback control is given in Section 2. We then describe the basic idea of our cooling scheme in Section 3.1, followed by the discussion of two approaches in Sections 3.2 and 3.3 to construct the jump operator such that the dissipative process drives the system towards the ground state. Our cooling scheme is benchmarked in Section 4, where the two approaches are applied to different models, including the one-dimensional Rice-Mele model [see Section 4.1] and the two-dimensional Haldane model [see Section 4.2]. A summary of the main results is given in Section 5 to conclude.

## 2 Markovian feedback control

Let us briefly recapitulate the idea of Markovian feedback control [31–33]. Suppose we perform a continuous measurement of the observable $M$ on a system described by the Hamiltonian $H$. The system dynamics is then governed by the stochastic master equation [31–33] ($\hbar = 1$ hereafter),

$$d\rho_c = -i[H, \rho_c]dt + \mathcal{D}[M]\rho_c dt + \mathcal{H}[M]\rho_c dW, \tag{1}$$

where $\rho_c$ denotes the quantum state (density matrix) conditioned on the measurement result, with nonlinear superoperators

$$\mathcal{D}[M]\rho := M\rho M^\dagger - \frac{1}{2}\left(M^\dagger M\rho + \rho M^\dagger M\right), \tag{2}$$

$$\mathcal{H}[M]\rho := M\rho + \rho M^\dagger - \mathrm{Tr}[\left(M + M^\dagger\right)\rho]\rho, \tag{3}$$

which describe the dissipation induced by the measurement, with $dW$ the standard Wiener increment with mean zero and variance $dt$. The measurement signal is given by

$$I_{\mathrm{hom}} = \mathrm{Tr}[\left(M + M^\dagger\right)\rho_c] + \xi(t), \tag{4}$$

with $\xi(t) = dW/dt$. By using the information obtained from the measurements, one can introduce feedback control to the system such as to steer the system's dynamics for achieving desired effects.

Here we consider the so-called Markovian feedback scheme introduced by Wiseman and Milburn [31], where the unprocessed measurement signal is fed back to the system by coupling it to an observable $F$. That is, we are introducing a term $I_{\text{hom}}F$ to the system. For such a feedback, the delay time between the measurement and the application of the control field is assumed to be negligible compared to the typical timescales of the system. For instance, in cold atom experiments, the typical time scales (such as tunneling time) are on the order of milliseconds [91]. Hence, a control on the higher kHz scale is sufficient, which can be achieved easily by using digital signal processors.

According to Markovian feedback control theory [32, 33], the combined action of measurement and feedback results in an effective dissipative process described by the feedback-modified stochastic master equation

$$d\rho_c = -i[H + H_{\text{fb}}, \rho_c]dt + \mathcal{D}[C]\rho_c dt + \mathcal{H}[C]\rho_c dW, \tag{5}$$

with the quantum jump operator

$$C = M - iF, \tag{6}$$

and feedback-induced term $H_{\text{fb}} = \frac{1}{2}(M^\dagger F + FM)$. By taking the ensemble average of the possible measurement outcomes, we arrive at the Wiseman-Milburn master equation

$$\frac{d\rho}{dt} = -i[H + H_{\text{fb}}, \rho] + \mathcal{D}[C]\rho. \tag{7}$$

Suppose the measurement strength is $\gamma$, i.e., $M \propto \sqrt{\gamma}$. For the feedback, we assume $F \propto \sqrt{\gamma}$ so that the jump operator $C \propto \sqrt{\gamma}$, and the feedback-induced term $H_{\text{fb}}$ is on the order of $\gamma$. In this work, we consider weak measurement, with $\gamma$ small enough such that $H_{\text{fb}}$ has negligible impact as compared to the system Hamiltonian $H$. Excluding the impact of $H_{\text{fb}}$, the steady state of Eq. (7) is given by

$$\mathcal{L}\rho \equiv -i[H, \rho] + \mathcal{D}[C]\rho = 0. \tag{8}$$

For weak measurement, the steady state is well approximated by a mixture of eigenstates of the system, with the weight dependent on the specific form of the jump operator $C$. In the following, we will discuss how to design $C$ to achieve a target steady state.

## 3 Constructing measurement and feedback operators

### 3.1 Basic idea

We consider discrete tight-binding models for fermionic ultracold atoms in an optical lattice at half filling. We focus on systems whose single-particle energy spectrum is characterized by two energy bands with non-trivial topological properties, such that the ground state $|g\rangle$ at half filling is a topological insulator. Our goal is to design the jump operator $C$, i.e., the underlying measurement and feedback, such that the effective dissipative dynamics drives the system towards its many-body ground state $|g\rangle$. The system (8) has a pure steady state if the effective Hamiltonian $H_{\text{eff}} = H - iC^\dagger C/2$ and the collapse operator $C$ have a common eigenstate [92], which is then the steady state. Hence, the ground state $|g\rangle$ will be a steady state of the system if it is a dark state of the jump operator $C$, i.e.,

$$C|g\rangle = 0. \tag{9}$$

Therefore, we need to construct a jump operator which satisfies the condition (9). It is intuitive that such a dissipation process will drive the system towards the ground state, since the coupling between the ground state and other eigenstates is unidirectional, i.e., the transfer

from other eigenstates to the ground state is allowed, while the reverse channel is blocked, as indicated by (9).

For the cooling scheme to be successful, the ground state should be the *unique* steady state of the system. Although this property in general depends on the details of the model considered, a general feature that can yield multiple steady states, spoiling the state preparation protocol, is degeneracy. For instance, consider two degenerate many-body eigenstates $|\psi\rangle$ and $|\psi'\rangle$ that are mapped by the jump operator to the same state $|\tilde{\psi}\rangle = C|\psi\rangle = C|\psi'\rangle$. Then, the superposition $|\psi_-\rangle = (|\psi\rangle - |\psi'\rangle)/\sqrt{2}$ will be both an eigenstate of $H_{\text{eff}}$ and of the jump operator with eigenvalue 0, $C|\psi_-\rangle = 0$. Therefore, $|\psi_-\rangle$ will also be a dark state like $|g\rangle$. Being a common eigenstate of $H_{\text{eff}}$ and $C$, it thus constitutes a steady state of the system. A key point for the success of our cooling scheme is, therefore, to ensure that there is no degeneracy in the system. In order to get rid of degeneracies in the example models, we will employ different strategies detailed in B.

In the following, we present two different choices for constructing a suitable jump operator $C$. We first discuss an *exact* construction, which is however challenging to implement experimentally. Starting from this, we then derive a second *approximate* construction for the purpose of enhancing the experimental feasibility.

## 3.2 Exact construction

We start by observing that, since the target state features all particles occupying the lower band only and filling all single-particle states therein, the jump operator $C$ must be able to deplete particles from the upper band and pump them to the lower band. It is further needed that particles can be pumped to any state in the lower band: this is guaranteed if the dissipative process provides non-zero transition matrix elements between any state in the upper band to any state in the lower band. Finally, we note that, while particles are pumped to the lower band, Pauli's exclusion principle will take care of inducing a uniform distribution of particles throughout the lower band, until the target state is eventually reached, by forbidding multiple occupancy. A jump operator complying with these conditions can be constructed as

$$C = \sqrt{\gamma}\, C_-^\dagger C_+ \,, \tag{10}$$

where $C_\pm$ destroys ($C_\pm^\dagger$, creates) a particle in states $|\pm\rangle = C_\pm^\dagger |0\rangle$ that have overlap with all states in the upper $(+)$ and in the lower $(-)$ band, respectively. In this way, the jump process transfers a particle from the upper band to the lower band. Concretely, we will choose $|\pm\rangle$ to be Wannier-like states for each band. Given the Bloch states $|k_\pm\rangle$ of the upper and lower band in a system with $N$ unit cells, the Wannier-like states are defined as

$$|\pm\rangle = \frac{1}{\sqrt{N}} \sum_{k \in \mathcal{B}} e^{i\varphi_{k_\pm}} |k_\pm\rangle \,, \tag{11}$$

where $\mathcal{B}$ is the first Brillouin zone and $e^{i\varphi_{k_\pm}}$ is a gauge factor associated to the Bloch state $|k_\pm\rangle$. By appropriate choice of $e^{i\varphi_{k_\pm}}$ we can obtain well-localized Wannier states. In terms of creation and annihilation operators for Bloch states $c_{k,\pm}$, such that $|k_\pm\rangle = c_{k,\pm}^\dagger |0\rangle$, the jump operator then reads

$$C = \frac{\sqrt{\gamma}}{N} \sum_{k,k' \in \mathcal{B}} e^{i(\varphi_{k'_+} - \varphi_{k_-})} c_{k,-}^\dagger c_{k',+} \,, \tag{12}$$

with $\gamma$ the measurement strength. It does indeed have a matrix element connecting any state in the upper band to any states in the lower band, as desired.

Within the Markovian feedback formalism, the collapse operator $C$ is related to the measurement operator $M$ and feedback operator $F$ via $C = M - iF$ in Eq. (6), where both $M$ and

$F$ are required to be Hermitian. Combining this with its Hermitian conjugate $C^\dagger = M + iF$, we obtain the explicit expressions: $M = (C + C^\dagger)/2$ and $F = i(C - C^\dagger)/2$. From the jump operator $C$ given in Eq. (12), the corresponding measurement and feedback operators can be identified as

$$M = \frac{\sqrt{\gamma}}{2N} \sum_{k,k' \in \mathcal{B}} \left( e^{i(\varphi_{k'_+} - \varphi_{k_-})} c^\dagger_{k,-} c_{k',+} - e^{i(-\varphi_{k'_+} + \varphi_{k_-})} c_{k,-} c^\dagger_{k',+} \right), \tag{13}$$

$$F = \frac{i\sqrt{\gamma}}{2N} \sum_{k,k' \in \mathcal{B}} \left( e^{i(\varphi_{k'_+} - \varphi_{k_-})} c^\dagger_{k,-} c_{k',+} + e^{i(-\varphi_{k'_+} + \varphi_{k_-})} c_{k,-} c^\dagger_{k',+} \right). \tag{14}$$

Even though the Wannier states can be chosen to be localized in real space (exponentially for topologically trivial bands and like a power-law for topologically non-trivial ones), they still span over the whole lattice. This may constitute a challenge for the experimental implementation of this approach, and motivates the search for a less demanding strategy, which we develop in the next section. For the exact construction of collapse operator, we set the gauge factor to be 0, since the localization of the Wannier states should not influence our cooling scheme in this case.

## 3.3 Approximate construction

In experiments, the measurement of on-site population in Eq. (13) can be implemented via homodyne detection of the off-resonant scattering of structured probe light from the atoms [93–95] and the feedback operator in Eq. (14) with a complex tunneling can be realized by accelerating the lattice [7, 10, 88, 96]. In the *exact* scheme, the feedback operator $F$ is typically *non-local* in real space, reflecting the delocalized nature of the optimal jump operator $C$. Implementing such non-local operations directly is a significant experimental challenge with current technologies, as it would require coherent control over spatially separated sites. While one could envision engineered feedback using global fields shaped by spatial light modulators or programmable optical potentials, these approaches are still limited in resolution and scalability. Thus, the exact scheme serves primarily as a theoretical benchmark that demonstrates the best possible performance under idealized feedback.

In order to favour experimental realizations, we develop a second approach in which the measurement and feedback operators are strictly constrained to have support on only a few neighbouring lattice sites. The idea is based on the fact that, by adjusting the gauge factors $e^{i\varphi_{k\pm}}$ in Eq. (12), the Wannier states can be chosen to be well localized in space, see more details in C. This approximate scheme is designed to preserve good performance while remaining implementable with current or near-term capabilities. In particular, the feedback operator $F$ will typically correspond, in this case, to local tunneling Hamiltonians acting within a small spatial region (e.g., a unit cell), with complex amplitudes, which could be implemented experimentally using techniques such as Floquet engineering [7, 10]. We start by considering a jump operator $C_\ell$ entirely localized in the $\ell$-th unit cell. We further use labels $A$ and $B$ for the two inequivalent sites in the unit cell, and $a_{\ell,A}$ ($a_{\ell,B}$) annihilates particles at site $A$ ($B$) in the $\ell$-th unit cell. We then choose an *ansatz* for the constrained jump operator of the form

$$C_\ell = \sqrt{\gamma} \, b^\dagger_{\ell,-} b_{\ell,+}, \tag{15}$$

with operators

$$b_{\ell,+} = \cos(\xi) a_{\ell,A} + \sin(\xi) a_{\ell,B}, \qquad b_{\ell,-} = -\sin(\xi) a_{\ell,A} + \cos(\xi) a_{\ell,B}. \tag{16}$$

These operators annihilate particles in states $|b_{\ell,\pm}\rangle = b^\dagger_{\ell,\pm} |0\rangle$, i.e.,

$$|b_{\ell,+}\rangle = \cos(\xi)|\ell,A\rangle + \sin(\xi)|\ell,B\rangle, \qquad |b_{\ell,-}\rangle = -\sin(\xi)|\ell,A\rangle + \cos(\xi)|\ell,B\rangle, \tag{17}$$

respectively, that are generic real superpositions of the states localized at $A$ and $B$ site parametrized in terms of an angle $\xi$. We choose real superpositions, such that they involve only one free parateter $\xi$. For the operator $C_\ell$ to successfully pump particles from the upper band to the lower band, we aim at selecting superposition states $|b_{\ell,\pm}\rangle$ such that $|b_{\ell,+}\rangle$ has maximal (minimal) overlap with the upper (lower) band, while $|b_{\ell,-}\rangle$ has maximal (minimal) overlap with the lower (upper) band. To find such states (the optimal angle $\xi$) we resort to analytical and numerical optimization schemes.

In the applications treated in the following, we will also consider jump operators constrained to a larger number of sites. Given a set $\mathcal{S}_n$ of $n$ sites to which the jump operator $C_{\mathcal{S}_n} = (b_-^{\mathcal{S}_n})^\dagger b_+^{\mathcal{S}_n}$ is constrained, we choose the *ansatz*

$$b_\pm^{\mathcal{S}_n} = \sum_{j \in \mathcal{S}_n} \beta_{\pm,j} a_j \,, \tag{18}$$

where the index $j$ indicates real-space coordinates. For the one-dimensional systems considered in the following, we can find optimal $|b_{\ell,\pm}\rangle$ analytically for $n = 2$. For larger $n$ and in other applications, we will find optimal coefficients $\beta_{\pm,j}$ through numerical optimization, minimizing the overlap of $|b_-^{\mathcal{S}_n}\rangle$ ($|b_+^{\mathcal{S}_n}\rangle$) with the upper (lower) band.

# 4 Application to paradigmatic models

We will now characterize the performance of the exact and approximate approaches for the preparation of the ground state for two different classes of models. We start from one-dimensional models, the Su-Schrieffer-Heeger [97] and the Rice-Mele model [98], respectively, see Section 4.1. Their one-dimensional band-structures (at fixed parameters) are not characterized by Chern numbers. However, they provide a minimal proof-of-principle scenario that allows us to test the scheme proposed above. In a second step, we investigate the two-dimensional Haldane model [2] both in its topologically trivial and non-trivial regime, see Section 4.2.

For the aforementioned models of moderate size, the steady state of the system is obtained by numerically solving Eq. (8) through exact diagonalization or via time evolution for a time sufficiently long to ensure that the system has reached a steady state [Section 4.2.1]. Note that the term $H_{\mathrm{fb}}$ is neglected throughout the paper, as justified in Appendix A. To quantify the performance of our cooling scheme, we calculate the fidelity between the steady state $\rho_{\mathrm{ss}}$ and the ground state $|g\rangle$ of the system, which is defined as

$$\mathcal{F} = \sqrt{\langle g| \rho_{\mathrm{ss}} |g\rangle} \,, \tag{19}$$

and takes values $0 \leq \mathcal{F} \leq 1$. A larger fidelity implies a better performance of our scheme.

Furthermore, in order to treat larger systems, we derive kinetic equations of motion using a mean-field approach for the case of the Haldane model [Section 4.2.2].

## 4.1 One-dimensional systems: The Rice-Mele and Su-Schrieffer-Heeger models

Our first example is the Rice-Mele model [98, 99], which describes a one-dimensional (1D) tight-binding chain with staggered hopping parameters and staggered on-site potentials [see sketch in Fig. 1(a)]. The Hamiltonian reads

$$H = \sum_l \Big[ E(n_{l,A} - n_{l,B}) - \big(J_1 a_{l,A}^\dagger a_{l,B} + J_2 a_{l,B}^\dagger a_{l+1,A} + \mathrm{h.c.}\big) \Big] \,, \tag{20}$$

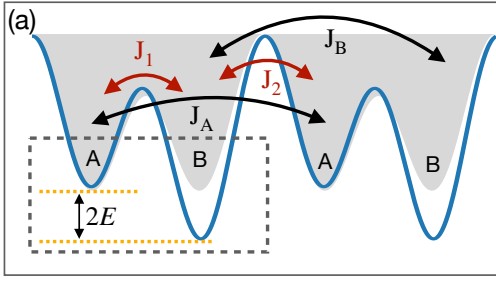 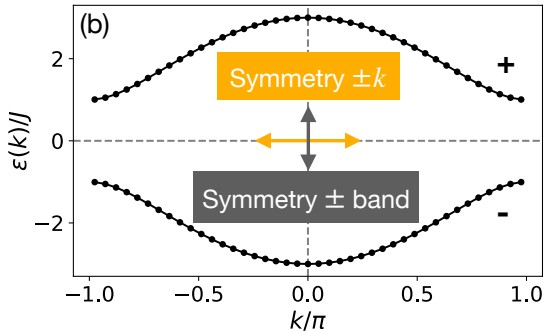

Figure 1: (a) A sketch of the Rice-Mele model in blue color, characterized by a staggered onsite potential, intracell hopping amplitude $J_1$, and intercell hopping amplitude $J_2$. $J_A$ and $J_B$ denote the next nearest neighbour hopping amplitudes. The potential can be realized by a superlattice, Eq. (21), in the experiments. The black dashed line indicates a unit cell. For $E = 0$ one obtains the Su-Schrieffer-Heeger model as a filled gray color, which is characterized by a symmetric double-well system. (b) Single-particle energy spectrum for the SSH model with periodic boundary condition, i.e., Eq. (24) for $E = 0$ and $J_1 = 2J_2 = 2J$. The symmetries related to $\pm k$ and between upper(+)/lower (-) bands are shown.

with $n_{l,\nu} = a^\dagger_{l,\nu} a_{l,\nu}$ being number operators for $\nu = A, B$. The chain consists of $N$ unit cells, each unit cell hosting two sites: one on sublattice $A$, one on sublattice $B$. Here, $J_1$ and $J_2$ denote the intracell and intercell nearest-neighbour tunnelling strength, and $E$ is the staggered onsite potential. This model can be realized with bichromatic ultracold atoms in optical lattices [100]. By using two counter propagating laser beams with period $d$ and $d/2$ respectively, one can obtain a superlattice with potential

$$V(x) = V_1 \sin^2\left(\pi \frac{x}{d}\right) + V_2 \sin^2\left(2\pi \frac{x}{d} + \theta\right), \tag{21}$$

where $\theta$ is the phase difference between these two different sublattices, and $V_1$ and $V_2$ are the lattice depths for the wider and narrower lattice, respectively.

For a translationally invariant chain with periodic boundary conditions (PBC), transforming the Hamiltonian to momentum space via $a^\dagger_{k,A/B} = \frac{1}{\sqrt{N}} \sum_\ell a^\dagger_{\ell,A/B} e^{ik\ell}$, where $k = q\frac{2\pi}{N}$ with $q = -\frac{N-1}{2}, \dots, \frac{N-1}{2}$, yields

$$H = \sum_{k \in \mathcal{B}} (a^\dagger_{k,A}, a^\dagger_{k,B}) \mathfrak{h}(k) \begin{pmatrix} a_{k,A} \\ a_{k,B} \end{pmatrix}, \tag{22}$$

characterized by the single-particle Hamiltonian

$$\mathfrak{h}(k) = \begin{pmatrix} E & -J_1 - J_2 e^{ik} \\ -J_1 - J_2 e^{-ik} & -E \end{pmatrix}, \tag{23}$$

whose eigenenergies read

$$\varepsilon(k) = \pm\sqrt{E^2 + (J_1 + J_2 \cos k)^2 + (J_2 \sin k)^2}, \tag{24}$$

with the spectrum shown in Fig. 1(b) for $E = 0$ and $J_1 = 2J_2$. The single-particle spectrum (24) has a symmetry with respect to $\pm k$ and a symmetry between upper and lower band. The former results from time-reversal invariance and the latter is a consequence of chiral symmetry, i.e., $\sigma_z \mathfrak{h}(k) \sigma_z = -\mathfrak{h}(k)$, with $\sigma_z$ being the Pauli matrix. Due to these symmetries, the many-body spectrum will be degenerate. For an effective cooling scheme, we need to lift the degeneracies

in the half filling spectrum, which could be realized in different ways, such as by introducing a weak nearest-neighbour (NN) interaction under open boundary condition (OBC), or by introducing next-nearest-neighbour (NNN) tunneling under OBC, see more in B.

### 4.1.1 Feedback cooling of the SSH chain

For $E = 0$ in the Hamiltonian (20), the Rice-Mele model reduces to the Su-Schrieffer-Heeger (SSH) model [97,99]. The Su-Schrieffer-Heeger (SSH) model shows a topological phase transition at $J_1/J_2 = 1$, separating a topologically trivial phase for $J_1/J_2 > 1$ from a non-trivial one for $J_1/J_2 < 1$, which is characterized by a non-trivial winding of the Berry phase through the Brillouin zone. To study the effectiveness of the exact method, we first consider open SSH chains of moderate size ($N = 1 \sim 4$) at half filling that allow us to numerically solve the steady-state equation $\mathcal{L}\rho = 0$, with the superoperator $\mathcal{L}$ given in Eq. (8) and jump operator $C$ of Eq. (10). To lift degeneracies in the many-body spectrum (see Section 3.1), we further introduce a weak nearest-neighbour interaction

$$\hat{H}_I = U \sum_\ell \left( n_{\ell-1,B} n_{\ell,A} + n_{\ell,A} n_{\ell+1,B} \right). \tag{25}$$

We solve Eq. (8) numerically using the toolbox QuTiP [101] in Python to find the steady state. As expected by construction, the exact method always yields the desired state $|g\rangle$ with close-to-unity fidelity between the steady state and the ground state for all system sizes investigated numerically, both in the topological ($J_1/J_2 < 1$) and trivial ($J_1/J_2 > 1$) phase.

Considering now the approximate method, we construct analytically optimal states $|b_{\ell,\pm}\rangle$ of Eq. (16) which are used to define the jump operator $C_\ell$ of Eq. (15) localized in the $\ell$-th unit cell. In particular, we search for states such that $|b_{\ell,-}\rangle$ ($|b_{\ell,+}\rangle$) has minimal overlap with the upper (lower) band. The eigenvectors of the single-particle momentum-space Hamiltonian (23) for $E = 0$ can be written as follows,

$$|k_\pm\rangle = \frac{1}{\sqrt{2}} \begin{pmatrix} \pm e^{i\phi_k} \\ 1 \end{pmatrix}, \qquad e^{i\phi_k} = \frac{-J_1 - J_2 e^{ik}}{|-J_1 - J_2 e^{ik}|}. \tag{26}$$

The squared overlap of state $|k_\pm\rangle$ with $|b_{\ell,-}\rangle$ is thus

$$|\langle k_\pm | b_{\ell,-}\rangle|^2 = \frac{1}{2N}\left[ 1 \mp \sin(2\xi)\cos(\phi_k) \right]. \tag{27}$$

Hence, the total overlap of the state $|b_{\ell,-}\rangle$ with the upper (+) and lower (−) band, shall be defined by the probability

$$\mathcal{P}_\pm(\xi) = \sum_{k \in \mathcal{B}} |\langle k_\pm | b_{\ell,-}\rangle|^2 = \frac{1}{2} \mp \frac{\sin(2\xi)}{2N} \sum_{k \in \mathcal{B}} \cos(\phi_k) \tag{28}$$

$$= \frac{1}{2} \mp \frac{\sin(2\xi)}{2N} s. \tag{29}$$

For convenience, in Eq. (29) we defined $s \equiv \sum_{k \in \mathcal{B}} \cos(\phi_k)$, which is always negative for a large system with $J_1 \neq 0$ and vanishes for $J_1 = 0$, since the $k$ values are equally spread over the first Brillouin zone.

Considering first $N = 1$ for simplicity, we have $s = -1$. In this case, the minimal (maximal) overlap of $|b_{\ell,-}\rangle$ with the upper (lower) band is attained at $\xi = -\pi/4$, leading to states

$$|b_{\ell,\pm}\rangle = \frac{1}{\sqrt{2}}(|\ell,A\rangle \mp |\ell,B\rangle). \tag{30}$$

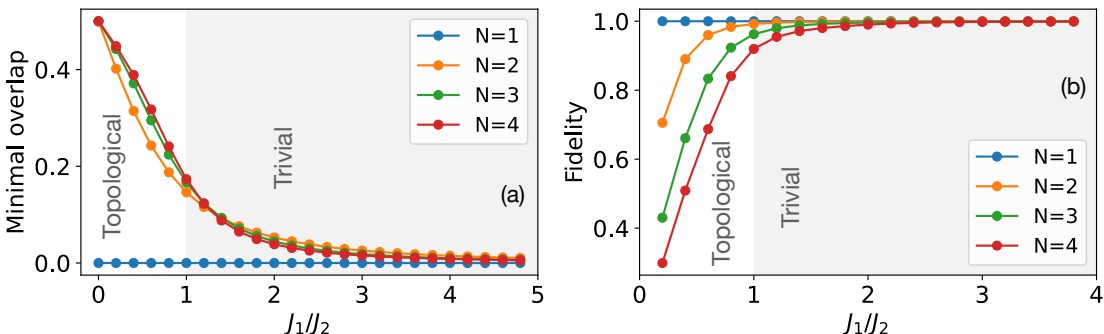

Figure 2: The approximate approach for the SSH model. The gray shaded background color indicates the trivial phase and the white background color indicates the topological phase. (a) The minimal overlap of Eq. (28) between the state $|b_{\ell,-}\rangle$ of lower band given by Eq. (30) and the upper band $|k_+\rangle$ for different small systems with $N$ unit cells as a function of $J_1/J_2$. (b) Fidelity defined in Eq. (19) as a function of $J_1/J_2$ with open boundary conditions for different small systems with $N$ unit cells. The steady state $\rho_{ss}$ is obtained by solving Eq. (8) with the jump operator constrained to two sites within one unit cell given by Eq. (31). The measurement strength is set to $\gamma = 0.0001J$ with $J$ the energy unit.

The orthogonality between $|k_+\rangle$ and $|k_-\rangle$ and between $|b_{\ell,+}\rangle$ and $|b_{\ell,-}\rangle$ then guarantees that $|\langle k_+|b_{\ell,-}\rangle|^2 = |\langle k_-|b_{\ell,+}\rangle|^2$, such that $|b_{\ell,+}\rangle$ has minimal overlap with the lower band. We discuss below that this construction works well numerically also for larger $N$ for $J_1/J_2 \gg 1$, while the regime $J_1/J_2 \ll 1$ can be treated as well by a shift of $|b_{\ell,\pm}\rangle$ by one lattice site. With this choice for the states $|b_{\ell,\pm}\rangle$, the jump operator $C_\ell$ of Eq. (15) is determined as

$$C_\ell = \frac{\sqrt{\gamma}}{2}\left(n_{\ell,A} - n_{\ell,B}\right) + \frac{\sqrt{\gamma}}{2}\left(a_{\ell,B}^\dagger a_{\ell,A} - a_{\ell,A}^\dagger a_{\ell,B}\right). \tag{31}$$

The corresponding measurement and feedback operators thus read

$$M_\ell = \frac{\sqrt{\gamma}}{2}\left(n_{\ell,A} - n_{\ell,B}\right), \qquad F_\ell = -i\frac{\sqrt{\gamma}}{2}\left(a_{\ell,A}^\dagger a_{\ell,B} - a_{\ell,B}^\dagger a_{\ell,A}\right). \tag{32}$$

Here, $M_\ell$ measures the population imbalance between the two sites $A$ and $B$ in the $\ell$-th unit cell, and $F_\ell$ denotes a tunneling between these two sites with a complex amplitude. The measurement of on-site population can be implemented via homodyne detection of the off-resonant scattering of structured probe light from the atoms [93–95]. The tunneling with a complex amplitude can be realized by accelerating the lattice [7, 10, 88, 96].

Figure 2(a) shows the minimal overlap $\sum_k |\langle k_+|b_{\ell,-}\rangle|^2$ between the state $|b_{\ell,-}\rangle$ given in Eq. (30) and the whole upper band. Figure 2(b) shows instead the fidelity given by the approximate cooling protocol as a function of $J_1/J_2$ for different system sizes ranging from 1 to 4 unit cells. As long as $J_1/J_2 > 1$ the fidelity is close to unity, while this is not the case for $J_1/J_2 < 1$ instead. This can easily be understood by considering that for $J_1 > J_2$ the system dimerizes, with the dimers localized in each unit cell. Maximally localized Wannier functions of the form of $|b_{\ell,\pm}\rangle$ [Eq. (30)] can thus be constructed, which are mainly localized within the $\ell$th unit cell (C). For $J_1 < J_2$, instead, the dimers straddle two neighbouring unit cells [99] and, in this case, the maximally localized Wannier functions also straddle two unit cells. Following this reasoning, successful cooling in the topological phase $J_1/J_2 < 1$ can also be achieved by choosing a collapse operator localized on neighbouring sites belonging to different unit cells. This can also be seen formally from Eq. (28): in the limit $J_2 = 0$, it holds that $\cos(\phi_k) = -1$ for any value of $k$, so the probability overlap $\mathcal{P}_\pm(\xi) = [1 \pm \sin(2\xi)]/2$ can reach 0 or 1 for an appropriate choice of $\xi$, while in the limit $J_1 = 0$ it becomes $1/2$ for $N > 1$ and is independent from $\xi$.

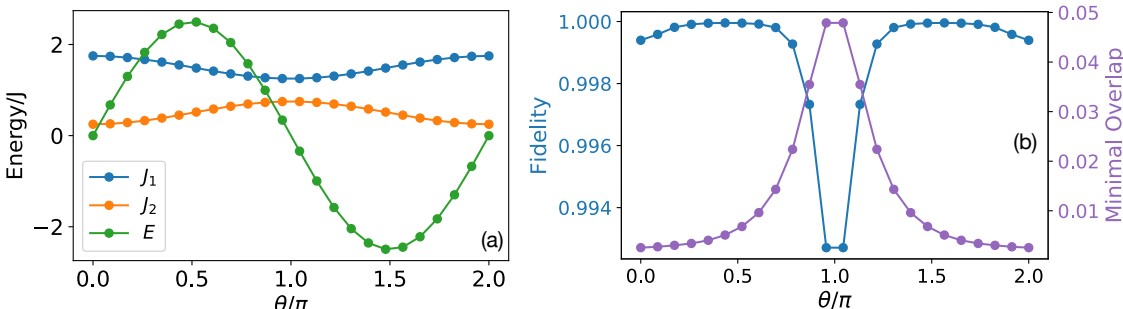

Figure 3: The approximate method for a Rice-Mele pumping cycle. (a) Hopping amplitudes $J_1$, $J_2$ and onsite potential $E$ as a function of $\theta$ given by Eq. (36). (b) The minimal overlap and the fidelity defined in Eq. (19) during the pumping cycle as a function of $\theta$ for a small system with 3 unit cells under open boundary condition. The steady state $\rho_{ss}$ is given by Eq. (8) and the jump operator constrained to two sites within one unit cell is given by Eq. (31). The measurement strength is set to $\gamma = 0.0001J$ with $J$ the energy unit. The steady state is obtained by solving Eq. (8).

### 4.1.2 Rice-Mele pumping cycle

We now construct the approximate jump operator $C_\ell$ for the Rice-Mele model of Eq. (20) with a non-zero staggered on-site potential $E \neq 0$. A state in the upper band can be written in the form

$$|k_+\rangle = \cos \frac{\chi_k}{2} |k,A\rangle + e^{-i\phi_k} \sin \frac{\chi_k}{2} |k,B\rangle \,, \tag{33}$$

where $\chi_k = 2\arctan\left[(\sqrt{E^2+J^2}-E)/J\right]$ with $-J_1-J_2 e^{ik} = J e^{i\phi_k}$. The squared overlap $\mathcal{P}_+(\xi)$ of $|b_{\ell,-}\rangle$ defined in Eq. (16) with the whole upper band is then

$$\mathcal{P}_+(\xi) = \frac{1}{2}\left\{1 - \frac{1}{N}\cos(2\xi)\sum_{k\in\mathcal{B}}\cos\chi_k - \frac{1}{N}\sin(2\xi)\sum_{k\in\mathcal{B}}\sin\chi_k\cos(\phi_k)\right\}. \tag{34}$$

If $\sum_{k\in\mathcal{B}}\cos\chi_k \neq 0$ and $2\xi \neq (n+1/2)\pi$ with integer $n$, the extremal point satisfies

$$\tan 2\xi = \frac{\sum_{k\in\mathcal{B}}\sin\chi_k\cos(\phi_k)}{\sum_{k\in\mathcal{B}}\cos\chi_k}\,. \tag{35}$$

With the approximate cooling operator determined by $\xi$, we study the cooling during a Rice-Mele pumping cycle. In this process, the parameters $E$, $J_1$ and $J_2$ in the Hamiltonian (20) are modulated periodically in time in a slow, adiabatic fashion. Through an appropriate parameter variation, the modulation pumps an integer number of particles along the chain which is determined by the Chern number of the valence band [102]. This so-called topological charge pump can be thought of as a 1D analog of the quantum Hall effect, where one spatial dimension is substituted by the temporal one [103]. We consider the following modulation of the on-site potential and hopping parameters [100, 104],

$$E(\theta) = \frac{5J}{2}\sin\theta\,, \qquad J_1(\theta) = \frac{J}{2}\left[3 + \frac{1}{2}\cos\theta\right]\,, \qquad J_2(\theta) = \frac{J}{2}\left(1 - \frac{1}{2}\cos\theta\right)\,, \tag{36}$$

as depicted in Fig. 3(a), with $J$ the energy unit. In an experiment, the modulation is achieved by varying the phase difference $\theta$ between the two sublattices, see Eq. (21). The cooling fidelity is reported in Fig. 3(b) as a function of $\theta$. In a large parameter regime, the deviation from unity is less than $10^{-3}$ and it remains always well below one percent, indicating that the approximate cooling protocol is reliable during the whole pumping cycle. The fidelity is slightly

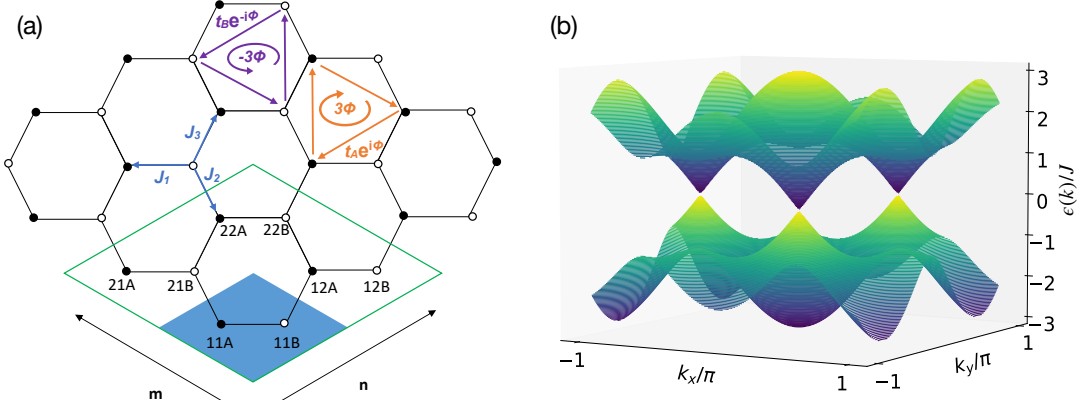

Figure 4: (a) Haldane model with hopping vectors in real space. The region in blue shading shows one unit cell with two sites $A$ and $B$. $J_1$, $J_2$ and $J_3$ are the nearest neighbour hopping amplitudes. The next nearest neighbour (NNN) hopping in a clockwise closed path with magnetic flux $3\phi$ enclosed is shown in orange color and the corresponding NNN hopping acquires a complex phase $e^{i\phi}$, and vice versa, the anticlockwise NNN hopping acquires a complex phase $e^{-i\phi}$. In order to enumerate the unit cells, we consider two directions with labels $m$ and $n$, which take integer values. Each lattice site is described by three parameters: $mn\sigma$, with $mn$ the index of unit cell and $\sigma = A, B$. (b) Single particle spectrum under periodic boundary condition for the Haldane model with $J_\lambda = J$, $t_A = t_B = 0.1J$, $\phi = \pi/2$, $\Delta = 0.52J$.

lower (although by a few parts in a thousand only) for values around $\theta \simeq \pi$. This effect can be explained by noting that at such values the on-site potential is close to zero. Non-zero values of $E$ indeed contribute in effectively reducing the hopping amplitude by bringing nearby sites off-resonant, thus favouring localized states that are well described by the single-cell ansatz (17) used for $|b_{\ell,\pm}\rangle$.

### 4.2 Two-dimensional topological insulator: The Haldane model

After having demonstrated the effectiveness of the cooling protocols in one-dimensional topological insulators, we now address the case of a two-dimensional system: the Haldane model of a Chern insulator [2]. This model substantially differs from previous examples, both because of its dimensionality and of its topological characterization. The Haldane model is defined on a honeycomb lattice with two sites ($A$ and $B$) per unit cell, as shown in Fig. 4(a). In order to enumerate the unit cells, we consider two directions with labels $m$ and $n$, which take integer values. Each lattice site is described by three parameters: $mn\sigma$, with $mn$ the index of unit cell and $\sigma = A, B$. The Hamiltonian is given by

$$
\begin{aligned}
H = - \sum_{\langle i,j \rangle_{\lambda=1,2,3}} J_\lambda \left( a_i^\dagger b_j + h.c. \right) &- \sum_{\langle\langle i,j \rangle\rangle_A} t_A \left( a_i^\dagger a_j e^{i\nu_{ij}\phi} + h.c. \right) \\
&- \sum_{\langle\langle i,j \rangle\rangle_B} t_B \left( b_i^\dagger b_j e^{i\nu_{ij}\phi} + h.c. \right) + \Delta \sum_i \left( a_i^\dagger a_i - b_i^\dagger b_i \right).
\end{aligned}
\tag{37}
$$

The summation over $\langle i, j \rangle_\lambda$ runs over (ordered) pairs of nearest neighbours (NN) with $\lambda = 1, 2, 3$, as shown in Fig. 4(a), while the summation over $\langle\langle i, j \rangle\rangle_{A/B}$ runs over next-nearest neighbours (NNN) between $AA$ or $BB$ sites. The complex phases $e^{i\nu_{ij}\phi}$ can be thought of as resulting from a magnetic field penetrating the lattice with zero net flux in each hexagon. The value of $\nu_{ij}$ is determined by the hopping directions with $\nu_{ij} = 1$ for clockwise hopping

and $\nu_{ij} = -1$ for counterclockwise hopping. The Haldane Hamiltonian thus features real-valued NN hopping parameters $(-J_\lambda)$ and complex NNN hopping parameters $(-t_\sigma e^{i\nu_{ij}\phi})$. We consider first the original Haldane model with $J_\lambda = J$ and $t_\sigma = t$, if not otherwise mentioned. The complex NNN coupling matrix elements break time-reversal symmetry and open gaps at both of the Dirac-type band-touching points, so that the resulting individual bands require topologically non-trivial properties characterized by Chern numbers $\pm 1$. In turn, the energy offset $\Delta$ breaks inversion symmetry. This term alone would open a topologically trivial band gap. If both terms are present, we find a competition between the two. By varying $\Delta$, the system enters a topological phase when $|\Delta| < |3\sqrt{3}t\sin\phi|$, which is characterized by the appearance of the chiral edge states [105].

### 4.2.1 Exact vs approximate method in small systems

To investigate feedback cooling in the Haldane model, we first focus on a system with $N = 4$ unit cells as depicted with the green line in Fig. 4(a). The steady state is obtained by solving Eq. (8) as done for the Rice-Mele model. Unwanted degeneracies arising from the symmetric single-particle spectrum shown in Fig. 4(b) are lifted by using open boundary conditions and adding a small nearest-neighbour interaction, see Eq. (25). The fidelity given by the exact method is shown in Fig. 5 for different values of the NNN tunneling phase $\phi$. As for the 1D case, the fidelity is very close to one for all parameter values, confirming the efficiency of the exact method. We further observe that the fidelity deviates slightly from one for smaller $\Delta$ and larger $\phi$: we attribute this discrepancy to the fact that, for the value $U = 0.001J$ of the interaction strength of Eq. (25) used, the system exhibits an increasing number of near-degeneracies in this regime, with decreasing level spacing as $\Delta$ decreases and $\phi$ increases. These near-degeneracies hinder the efficiency of the exact protocol through the phenomena discussed in Sec. 3.1.

We then test the approximate method by constructing constrained jump operators localized at two sites with indices 11$A$, 11$B$ in the first unit cell, and on four sites with indices 11$A$, 11$B$, 21$B$, 12$A$ indicated in Fig. 4(a). We can parameterize the wavefunctions localized at four sites

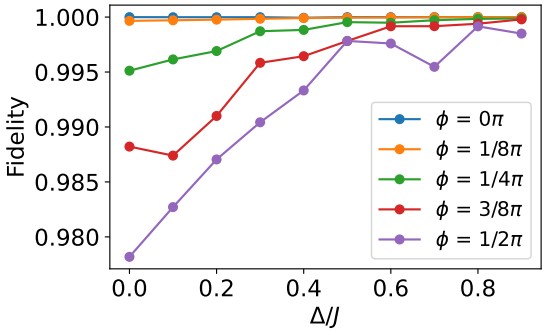

Figure 5: Fidelity between the steady state given by Eq. (8) with jump operator $C$ of Eq. (10) (for the exact approach) and the ground state of the Haldane model (37) for a small system with four unit cells as a function of the on-site potential $\Delta$ for different values of the complex hopping phase $\phi$. The steady state is obtained by solving Eq. (8). We use OBC with a small interaction (25) of strength $0.001J$ to break the degeneracy.



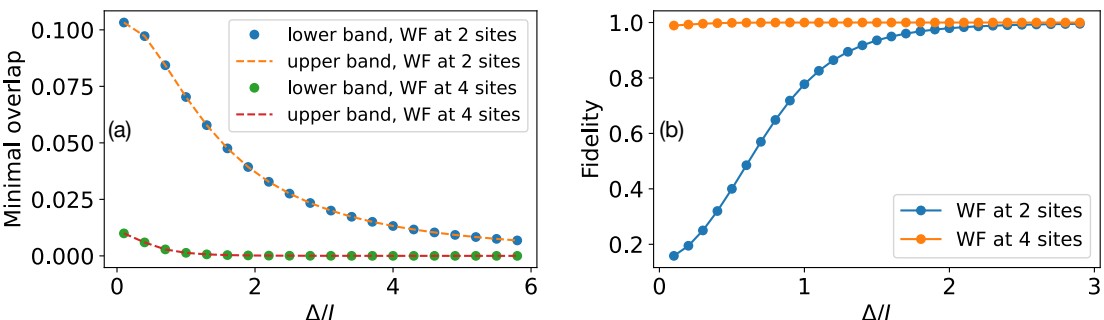

Figure 6: Approximate approach for a small system of the Haldane model with 4 unit cells as depicted with green line in Fig. 4(a). We consider the cases where the wavefunctions (WF) $|b_\pm^{\mathcal{S}_n}\rangle$ are localized at two sites with indices 11$A$, 11$B$ in the first unit cell, and on four sites with indices 11$A$, 11$B$, 21$B$, 12$A$ indicated in Fig. 4(a). We set $\phi = \pi/2$ and $t = 0.1J$, where the energy unit $J$ is the NN hopping amplitude. The parameterizations of $|b_\pm^{\mathcal{S}_n}\rangle$ are given by Eq. (17) for $n = 2$ and by Eq. (38) for $n = 4$. The parameters can be determined by minimizing the overlap of $|b_-^{\mathcal{S}_n}\rangle$ ($|b_+^{\mathcal{S}_n}\rangle$) with upper (lower) band, with the minimal overlaps shown in (a). (b) Fidelity between the steady state given by Eq. (8) and the ground state of the Haldane model for a small system with 4 unit cells as a function of the onsite potential $\Delta$ for $t = 0.1J$. The steady state is obtained by solving Eq. (8) with the jump operator $C_{\mathcal{S}_n} = (b_-^{\mathcal{S}_n})^\dagger b_+^{\mathcal{S}_n}$. The measurement strength is set to $\gamma = 0.0001J$.

with three parameters $\xi_1$, $\xi_2$ and $\xi_3$ in the following way,

$$\begin{aligned}
|b_\pm^{\mathcal{S}_4}\rangle = &\sin(\xi_1)\sin(\xi_2)\sin(\xi_3)|11A\rangle + \sin(\xi_1)\sin(\xi_2)\cos(\xi_3)|11B\rangle \\
&+ \sin(\xi_1)\cos(\xi_2)|21B\rangle + \cos(\xi_1)|12A\rangle .
\end{aligned} \tag{38}$$

Given the parametrization (38), the optimal parameters $\xi_1$, $\xi_2$ and $\xi_3$ can be determined by following a similar strategy as for the two-sites case, where we try to minimize the overlap of $|b_-^{\mathcal{S}_4}\rangle$ ($|b_+^{\mathcal{S}_4}\rangle$) with the upper (lower) band. The minimal overlaps found are shown in Fig. 6(a). For smaller on-site potential $\Delta$ (corresponding to the topological phase) the minimal overlaps are smaller for $|b_\pm^{\mathcal{S}_n}\rangle$ localized at four sites than at two sites. The amplitudes of the optimal wavefunction $|b_\pm^{\mathcal{S}_n}\rangle$ at different lattice sites are shown in Fig. 7. The resulting fidelity is shown in Fig. 6(b). For the two-site jump operator $C_{\mathcal{S}_2}$, the cooling is not efficient for small on-site potentials $\Delta/J \ll 1$, but the fidelity grows monotonically by increasing $\Delta$, eventually reaching $\mathcal{F} = 1$ for $\Delta/J \sim 3$. The situation strikingly improves when considering $C_{\mathcal{S}_4}$, localized on four sites. In this case the fidelity barely deviates from one at vanishing $\Delta$, and remains otherwise very close to one for all values of $\Delta$. For small on-site potentials, the 2D system is in a topological phase and it is not possible to construct maximally localized Wannier functions [105], which leads to poor cooling performance of the two-site algorithm. This differs significantly from the 1D SSH model 4.1.1, where cooling in the topological phase still works by simply shifting the operators $|b_{\ell,\pm}\rangle$, as the maximally localized Wannier functions are shifted in this case.

### 4.2.2 Cooling large systems: Mean-field approach

Having benchmarked the performance of both the exact and the approximate methods in small systems by exact numerical determination of the steady state, we now investigate cooling in the (non-interacting) Haldane model for larger systems by adopting a mean-field approach. In

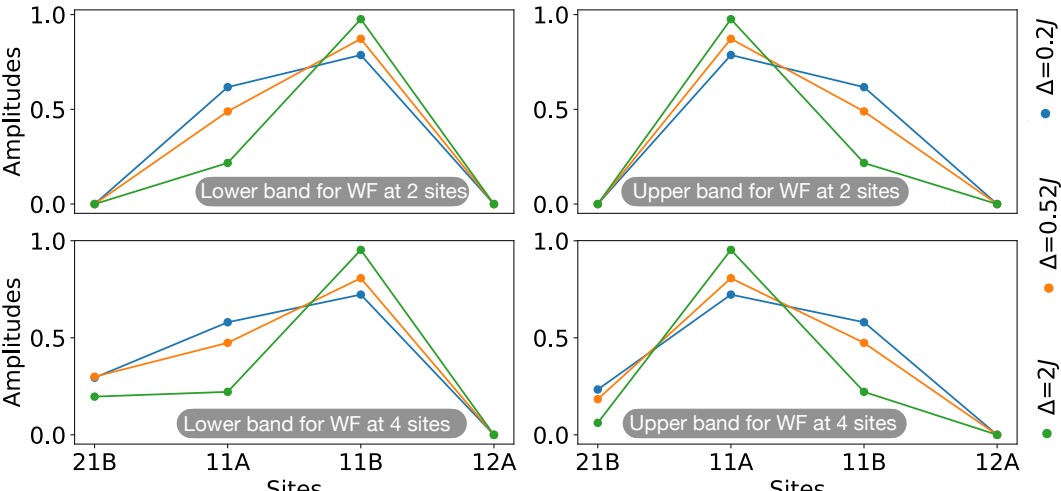

Figure 7: Amplitudes of the optimal wavefunctions $|b_{\ell=1,\pm}\rangle$ at two sites in the first unit cell with the ansatz given by Eq. (17) and $|b_{\pm}^{\mathcal{S}_4}\rangle$ at four sites with the ansatz given by Eq. (38). The different colors stand for different on-site potentials $\Delta$. We consider a small system of the Haldane model with 4 unit cells as depicted with green line in Fig. 4(a). We set $\phi = \pi/2$ and $t = 0.1J$ with $J$ the NN hopping amplitude. To get the optimal parameters we try to minimize the overlap of $|b_{-}^{\mathcal{S}_n}\rangle$ ($|b_{+}^{\mathcal{S}_n}\rangle$) with upper (lower) band. We choose the two sites with indices 11A, 11B in the first unit cell, and the four sites with indices 11A, 11B, 21B, 12A, as shown in Fig. 4(a).

particular, following Refs. [106, 107] we derive kinetic equations of motion for the mean occupations of the single-particle eigenstates (D). The resulting non-linear equations of motions read

$$\frac{d}{dt}\bar{n}_k(t) = \sum_q \Big( R_{kq}\bar{n}_q[1 - \bar{n}_k(t)] - R_{qk}\bar{n}_k(t)[1 - \bar{n}_q(t)] \Big), \tag{39}$$

where $\bar{n}_k(t) = \text{tr}[n_k\rho(t)]$ is the mean occupation number of the $k$-th single-particle eigenstate (ordered by increasing energy). These equations are obtained by neglecting non-trivial correlations, $\text{tr}[n_k n_q \rho(t)] \approx \bar{n}_k(t)\bar{n}_q(t)$, which is justified in the limit of large systems. The validity of the mean-field approximation for small systems is further studied in Section E in the Appendix, where the exact full-density-matrix approach is compared with mean-field approach. The transition rates $R_{kq}$ are obtained from the feedback master equation (7) after an additional rotating-wave-approximation (RWA), which is valid if the single-particle energy gaps and their differences are much larger than the measurement strength $\gamma$. The rates are derived from the feedback jump operator $C$ according to $R_{kq} = |C_{kq}|^2$.

In the case of the exact method, we see from Eq. (12) that the rates $R_{kq}$ are such that $R_{kq} = 1$ (in the dimensionless units used), if the eigenstates labeled by $k$ and $q$ belong to the lower and upper band, respectively, while $R_{kq} = 0$ otherwise. The mean-field equation (39) for the occupation of a state $k$ in the lower band then reduces to

$$\frac{d}{dt}\bar{n}_k(t) = [1 - \bar{n}_k(t)] \sum_{q \in B_+} \bar{n}_q(t), \tag{40}$$

where $B_+$ denotes the upper band. The steady state, $d\bar{n}_k(t)/dt = 0$, is then easily found to be $\bar{n}_k(t) = 1$, confirming that an exact preparation of the many-body ground state is indeed attained, featuring all states in the lower band occupied. We then investigate the performance of the approximate method by constraining the jump operator to different sets of sites $\mathcal{S}_n$ and

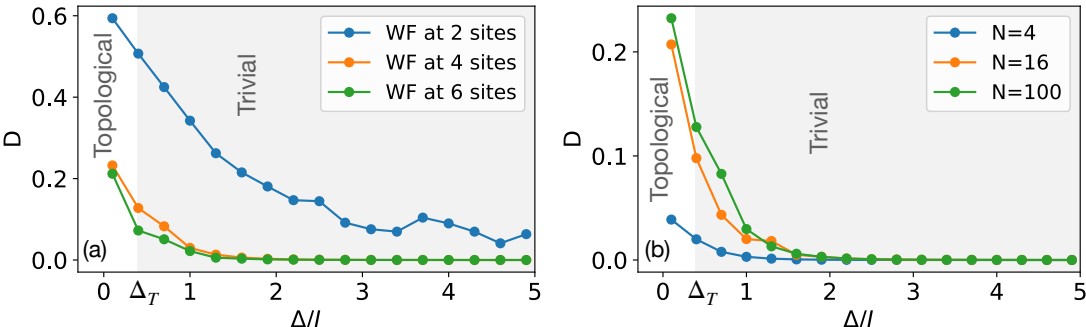

Figure 8: Efficiency of the approximate construction for the Haldane model. We plot the deviation given by (41) between the steady state of the mean-field equations (39) and the ground state as a function of on-site potential $\Delta$ for $\phi = \pi/2$. We consider slightly different NNN hoppings with $t_A = 0.1J$, $t_B = 0.05J$, where $t_A$ denotes the NNN hopping amplitude without phase factor for *AA* sites and $t_B$ denotes the NNN hopping amplitude without phase factor for *BB* sites. $\Delta_T$ denotes the topological phase transition. (a) We consider a large system with 100 unit cells arranged in $10 \times 10$, as shown in Fig. 4(a). The wavefunctions are constrained at two, four or six sites. The choice of $\mathcal{S}_2$ and $\mathcal{S}_4$ is same with Fig. 6. $\mathcal{S}_6$ is given by 11*A*, 11*B*, 21*B*, 12*A*, 22*A* and 22*B* as indicated in Fig. 4(a). (b) Deviation for different large systems with $N$ unit cells arranged in a manner such that $m$ and $n$ run from 1 to $\sqrt{N}$, as shown in Fig. 4(a). The wavefunctions are constrained at four sites with the choice of $\mathcal{S}_4$ same with (a).

solving the mean field equations (39) numerically in a system of $N = 100$ unit cells arranged in a manner such that $m$ and $n$ run from 1 to 10, as shown in Fig. 4(a). To ensure that the spectrum does not feature two identical level spacings (see D), such that the RWA can be assumed to be valid for a suitably small $\gamma$, we consider slightly different strength of the NNN hopping between two *A* sites, denoted with $t_A$, as compared to two *B* sites, denoted with $t_B$ (see Fig. 4(a)). The construction of the constrained jump operators is performed via numerical optimization as explained in Section 3.3. We quantify the deviation $D(\bar{\boldsymbol{n}})$ from the ground state by comparing the mean occupation numbers in the steady state, $\bar{\boldsymbol{n}} = \{\bar{n}_k\}_{k=1,\ldots,2N}$ with those in the ground state, $\bar{\boldsymbol{n}}^{(g)} = \{\bar{n}_k^{(g)}\}_{k=1,\ldots,2N}$, computing

$$D(\bar{\boldsymbol{n}}) = \sqrt{\frac{1}{2N} \sum_{k=1}^{2N} (\bar{n}_k - \bar{n}_k^{(g)})^2} \,. \tag{41}$$

The sets $\mathcal{S}_n$ considered are localized on $n = 2$, 4 and 6 sites. The choice of $\mathcal{S}_2$ and $\mathcal{S}_4$ is the same as in Fig. 6. $\mathcal{S}_6$ is given by 11*A*, 11*B*, 21*B*, 12*A*, 22*A* and 22*B* as indicated in Fig. 4(a). The Deviation $D$ obtained as a function of $\Delta$ is depicted in Fig. 8. We observe that the deviation is fairly large for $\mathcal{S}_2$ (blue curve), but it gets much closer to zero as the number of sites is increased, for $\mathcal{S}_4$ (orange curve) and $\mathcal{S}_6$ (green curve). In the latter cases, the deviation tends to zero for $\Delta/J > 1$, but deviates from zero for $\Delta/J < 1$.

The above effect can be understood in the light of the topological phase transition, thus interestingly linking the efficiency of the engineered cooling mechanism with the system's topological properties. Indeed, around $0 < \Delta_T < 1$, the system crosses the phase transition and enters the topological phase, where exponentially localized Wannier functions do not exist, such that the constrained ansatz states $|b_\pm^{\mathcal{S}_n}\rangle$ cannot reproduce the eigenstates to a satisfactory degree. To bring further evidence that this effect is indeed related to the topological phase

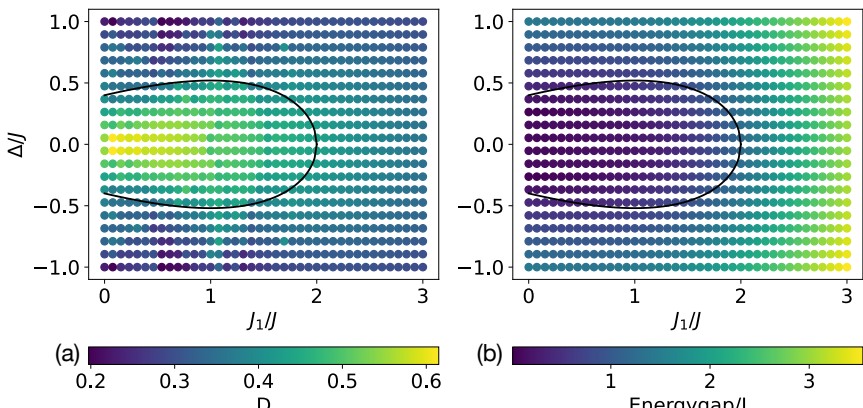

Figure 9: (a) Deviation between final state of the mean-field equations (39) and the ground state of the Haldane model for a large system with 16 unit cells arranged in the same way as Fig. 8. (b) The corresponding energy gaps under open boundary conditions. The approximate approach with wavefunctions localized at 2 sites is applied here. The parameters are set as $J_2 = J_3 = J$ and $t = 0.1J$. The black curves describe the topological transition, which is given by Eq. (42). The horizontal axis is $J_1$ and vertical axis is the on-site potential $\Delta$. The deviation $D$ given by (41) and the energy gap, which are functions of $J_1$ and $\Delta$, are represented with a color map.

transition, we show in Fig. 8(b) the deviation for different system sizes, again as a function of $\Delta$. The steepness of the curve at low $\Delta$ increases for increasing size, as expected, since it is more difficult to find Wannier functions localized at four sites in the topological phase. As an additional signature, we study how the cooling improves by entering a trivial dimerized phase, in which the tunneling strength $J_1$ along the direction $e_1$ is much larger than in other directions [see Fig. 4(a)]. For the case of a dimerization with changing $J_1$ and constant $J_2 = J_3 = J$ the topological phase transition is given by

$$\Delta \pm t \left( \frac{J_1}{J} + 2 \right) \sqrt{4 - \frac{J_1^2}{J^2}} = 0 \,. \tag{42}$$

The deviation for this case is reported in Fig. 9(a), and indeed becomes smaller for increasing $J_1$. This observation together with the results in Fig. 8 indicate that the deviation is related with the energy gap, so we plot the energy gaps of the Haldane model in Fig. 9(b) for the approximate approach with jump operators localized at two sites under open boundary condition. We observe that $D$ is smaller (larger) for a larger (smaller) energy gap, which indicates that the cooling performance is much better for a larger energy gap.

## 5  Conclusions

In summary, we have proposed a scheme to prepare topological insulator states for noninteracting fermionic atoms in an optical lattice through Markovian feedback control. Specifically, we considered topologically non-trivial two-band models at half-filling, and exploited continuous weak measurement and Markovian feedback to engineer a dissipative process which cools the system towards the ground state. This is achieved, in turn, by constructing a dissipative process that pumps particles from the upper band to the lower band, until the latter is filled. We further propose approximate variant schemes that can perform the same task with lower efficiency, when additional experimental constraints on the measurement and feedback apparatus are introduced. We have benchmarked these two approaches in several 1D and 2D

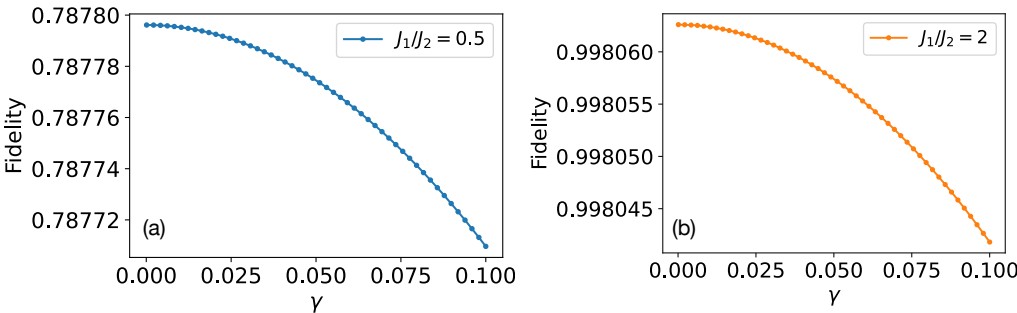

Figure 10: Fidelities for a small SSH model with 3 unit cells for different values of measurement strength $\gamma$ by considering two different topological regimes (a) topological phase and (b) trivial phase, for the approximate approach with Wannier function localized in one unit cell.

lattice models, namely for the Su-Schrieffer-Heeger model, the Rice-Mele model and the Haldane model. For moderate system sizes, we probed the steady state of the system by solving the feedback-modified master equation numerically. For large systems, we resorted to kinetic theory and compared the mean occupations of the single-particle eigenstates. The proposed exact cooling scheme is successful in all parameter regimes and for all models studied. The approximate methods, which involve a restriction of the measurement and feedback operations to small subsystems, give good performance for small systems or when the system eigenstates tend to be localized on few sites. While this makes this approach less effective in the topological phase of the 2D Haldane model for large systems, it still gives a satisfactory preparation of topological insulator states in the 1D models studied.

# Acknowledgments

**Funding information** This research was funded by the Deutsche Forschungsgemeinschaft (DFG, German Research Foundation) via the Research Unit FOR 5688 "Driven-dissipative many-body systems of ultracold atoms", project number 521530974, and by the Hainan Province Science and Technology Talent Innovation Project (Grant No. KJRC2023L05). F. P. acknowledges funding from the Deutsche Forschungsgemeinschaft (DFG, German Research Foundation) through the Emmy Noether Programme – project number 555842149.

# A   Impact of measurement strength on fidelity

For the simulations in the paper, we did not include the feedback term $H_{\text{fb}}$. To verify that this approximation does not alter the results significantly and to address the impact of the measurement strength $\gamma$ on the steady-state fidelity, we have added $H_{\text{fb}}$ to the system Hamiltonian and computed fidelities for a small SSH model with three unit cells, considering different values of $\gamma$ and two different topological regimes: $J_1/J_2 = 2$ and $J_1/J_2 = 0.5$, as shown in Fig. 10. The setup is the same as in Fig. 2 for the approximate approach. By slowly changing $\gamma$ from 0.0001 to 0.1, we observe that a smaller $\gamma$ gives slightly higher fidelity, but in both cases the influence of $\gamma$ in fidelity is on the order of $10^{-5}$, thus justifying the omission of the $H_{\text{fb}}$ term in the simulations.

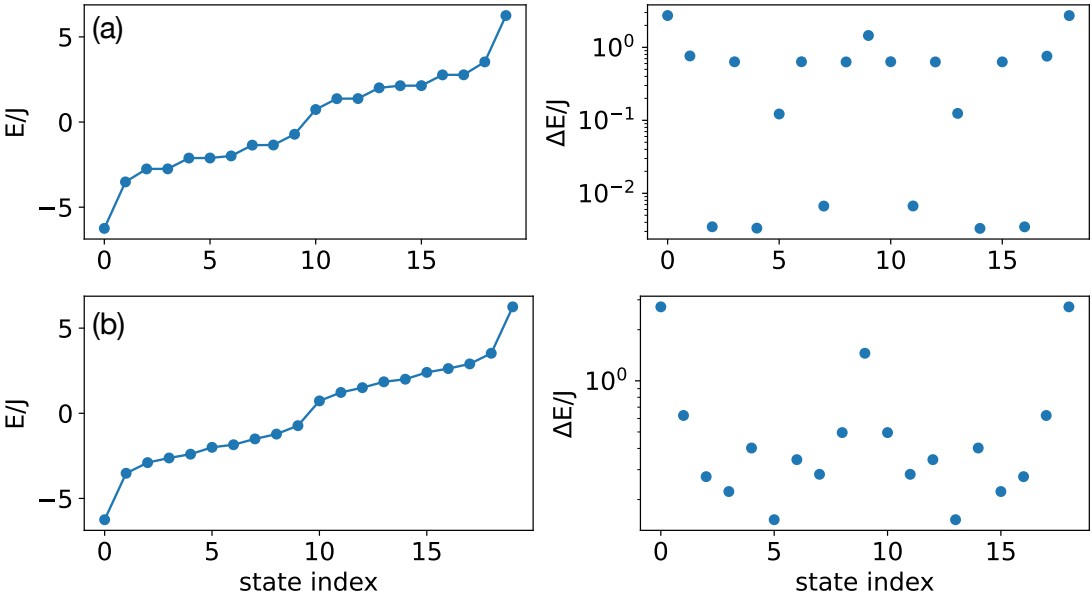

Figure 11: Half-filling spectrum for a system of 3 unit cells in SSH model with $J_1 = 2J$ and $J_2 = J$. The eigenenergies are sorted in an ascending way and enumerated along the x-axis. Figures on the left side are plots of the energy spectrum in linear scale. Figures on the right side are in semilog scale and describe the energy difference between neighboring points of the left plots. (a) Plots for the case of OBC with small interaction of strength 0.01J. (b) Plots for the case of OBC with NNN hopping strength 0.1J.

## B  Different methods to lift degeneracies of the many-body spectrum

The degeneracies in the many-body spectrum arise from the symmetric single-particle spectrum with respect to $\pm k$ and between the two bands, as shown in Fig. 1(b). We study different ways to lift the degeneracies.

(1) We introduce a small interaction between nearest neighbors with interaction Hamiltonian in Eq. (25), with $U$ denoting the interaction strength, and use open boundary conditions (OBC). For OBC $l$ runs from 1 to $N-1$ in Eq. (20). We plot the half-filling spectrum and the energy difference between neighbouring states in Fig. 11(a), where all the energy differences $\Delta E$ are non-zero, which means that the degeneracy is lifted by adding small nearest-neighbour interactions.

(2) We can introduce more tunneling coefficients, such as next nearest neighbor tunneling coefficients (NNN hopping). Similar as in (1), the half-filling spectrum and the energy differences are plotted in Fig. 11(b).

## C  Maximally localized Wannier function in SSH model

We recall that Wannier functions are used to construct the collapse operators in Eq. (12). The approximate approach, where the collapse operator is chosen to be localized at a few sites, is based on the fact that by choosing appropriate gauge factors it is possible to find well localized Wannier states in Eq. (12). Here, as an example, we try to construct maximally localized Wannier function $|W_-(l)\rangle$ in the tight binding SSH model with $J_1 = 2J_2$, for the lower

(—) band, where $l$ lists all unit cells. In order to find the appropriate gauge factor $e^{i\varphi_{k_-}}$, we perform either a single-band transformation following Ref. [108] or use the Kohn gauge [109]. In the Kohn gauge, the gauge factor can be chosed as

$$\varphi_{k_-} = -\arg(|k_-\rangle_1 + |k_-\rangle_2),\tag{C.1}$$

where $|k_-\rangle_1$ and $|k_-\rangle_2$ are the two components of the lower band eigenvector $|k_-\rangle$ in Eq. (26). The computed Wannier function is shown in Fig. 12, where we notice an exponential localization of the Wannier function in space. We can calculate the weight of the two sites $0A$ and $0B$ in the 0th unit cell, which is $|W_-(0A)|^2 + |W_-(0B)|^2 = 0.97$, where the Wannier function is normalized. This result implies that the approximate approach for SSH model should work very well for $J_1 = 2J_2$, which is consistent with the results in Fig. 2.

## D   Derivation of the mean-field equations

Since it is very demanding to calculate the steady state $\rho_{ss}$ numerically for large systems, we can adopt a mean-field description in which we rather study the time evolution of the mean occupation of single-particle eigenstates, following Refs. [106, 107]. To work out this mean-field description, we first need to recast the master Eq. (7) (with $H_{\text{fb}} = 0$) in a form which describes quantum jumps between single-particle eigenstates. This can be achieved, while maintaining Lindblad form, as follows. We first represent the collapse operator $C$ in the system's eigenbasis, $C = \sum_{k,q} C_{kq} a_k^\dagger a_q = \sum_{k,q} C_{kq} L_{kq}$, where $L_{kq} = a_k^\dagger a_q$ describes a quantum jump from single-particle eigenstate $|q\rangle$ to $|k\rangle$. In interaction picture, the master equation then reads as

$$\frac{d\rho}{dt} = \sum_{k,q,k',q'} C_{kq} C_{k'q'}^* e^{i(\omega_{kq} - \omega_{k'q'})t} \Big[ L_{kq} \rho L_{k'q'}^\dagger - \frac{1}{2} L_{k'q'}^\dagger L_{kq} \rho - \frac{1}{2} \rho L_{k'q'}^\dagger L_{kq} \Big],\tag{D.1}$$

where $\omega_{kq} = \epsilon_k - \epsilon_q$ is the energy difference between the single-particle levels $\epsilon_k$ and $\epsilon_q$. Note that the master equation in Eq. (D.1) is already in Lindblad form, as it is derived from Eq. (7), which preserves complete positivity. To further derive the kinetic equations, we next adopt the rotating-wave (secular) approximation, which amounts to neglect oscillating terms in Eq. (D.1). This is justified only in the regime where the difference in level spacings $\omega_{kq} - \omega_{k'q'}$ is much larger than the measurement strength $\gamma$, $|\omega_{kq} - \omega_{k'q'}| \gg \gamma$. To ensure that this condition can be met, in our numerical studies we use the methods described in B to prevent the emergence of identical level spacings in the single-particle spectrum. In rotating-wave approximation, Eq. (D.1) then becomes

$$\frac{d\rho}{dt} = \sum_{k,q} R_{kq} \Big[ L_{kq} \rho L_{kq}^\dagger - \frac{1}{2} L_{kq}^\dagger L_{kq} \rho - \frac{1}{2} \rho L_{kq}^\dagger L_{kq} \Big],\tag{D.2}$$

where we introduced the effective quantum jump rates $R_{kq} = |C_{kq}|^2$. From Eq. (D.2), one can derive an equation ruling the evolution of the single-particle occupations $\bar{n}_k = \text{tr}(\rho n_k)$. The latter will also depend on two-particle correlations, $\text{tr}(\rho n_k n_q)$, initiating a hierarchy of equations for the $n$-particle correlation functions [107]. The hierarchy can be truncated in a mean-field-like approximation by assuming the factorization of two-particle correlations, $\text{tr}(\rho n_k n_q) \approx \bar{n}_k \bar{n}_q$ [107]. This procedure yields non-linear mean-field equations for the single-particle occupations, reading as

$$\frac{d}{dt} \bar{n}_k(t) \approx \sum_q \big\{ R_{kq} \bar{n}_q(t)[1 - \bar{n}_k(t)] - R_{qk} \bar{n}_k(t)[1 - \bar{n}_q(t)] \big\}.\tag{D.3}$$

In the main text, we studied the steady-state occupations given by these mean-field equations, satisfying $\frac{d}{dt}\bar{n}_k(t) = 0$, which have been found numerically both via long-time propagation and through a nonlinear-equation solver.

# E   Validity of the mean-field approximation

The validity of the mean-field approximation—namely the factorization $\langle n_k n_q \rangle \approx \langle n_k \rangle \langle n_q \rangle$ used in deriving the kinetic equations—is a key assumption for applying our approach to large systems. To assess the reliability of this approximation, we compare the exact results with the mean field results for a small SSH system of four unit cells with $J_1/J_2 = 2$. We first calculate the steady state $\rho_{ss}$ of the system by solving the master equation. Via $\rho_{ss}$ we compute $\langle n_k n_q \rangle = \text{tr}[\rho_{ss} n_k n_q]$ and $\langle n_k \rangle \langle n_q \rangle$, as shown in Fig. 13. We observe that $|\langle n_k n_q \rangle - \langle n_k \rangle \langle n_q \rangle|$ is very small, thus justifying the validity of mean-field approximation.

To further substantiate our approach, we have compared the deviation measure $D$ calculated via both exact diagonalization (ED) and mean-field theory [Fig. 14(a)]. The observed close agreement between these methods strongly supports the reliability of our mean-field approximation.

Furthermore, we have investigated the relationship between the deviation measure $D$ and the fidelity $\mathcal{F}$. Remarkably, as demonstrated in Fig. 14(b), $1-D$ exhibits behavior qualitatively similar to the fidelity $\mathcal{F}$, suggesting that $1-D$ serves as an excellent proxy for $\mathcal{F}$ in our system.

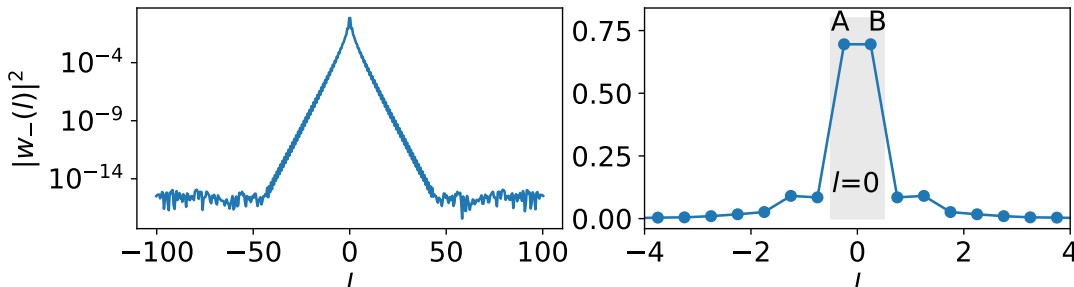

Figure 12: MLWF in tight binding model for SSH model with 201 unit cells with $J_1 = 2J_2$, for the lower ($-$) band. The index $l$ lists all unit cells from -100 to 100. The gray region shows the 0th unit cell with $A$ and $B$ sites. In this case the Wannier function is exponentially localized. Plots on the left side are in semilog scale. Plots on the right side are in linear scale and zoom on the peak of the Wannier functions.

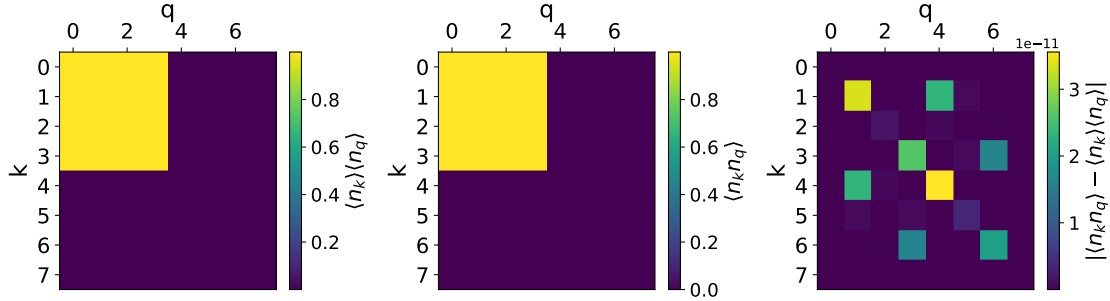

Figure 13: Validity of mean-field approximation for a small SSH system of 4 unitcells with $J_1/J_2 = 2$: comparison between $\langle n_k n_q \rangle$ and $\langle n_k \rangle \langle n_q \rangle$.

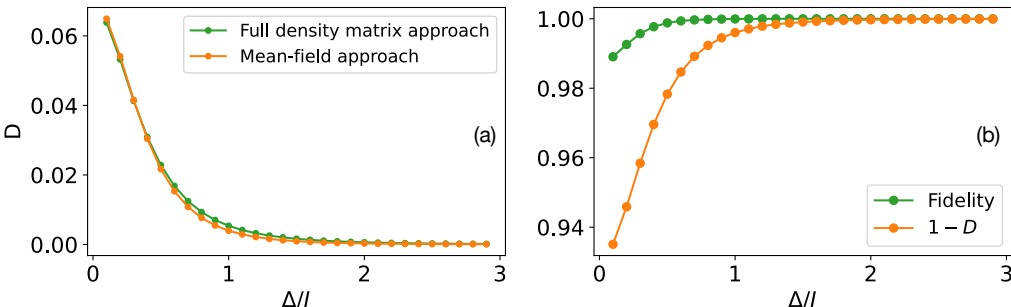

Figure 14: (a) Comparison between the values of the deviation $D(\bar{n})$, defined in Eq. (41) of the manuscript, obtained using two different methods: the full steady-state density matrix and the mean-field approach. (b) Direct comparison between fidelity and $1 - D$. These studies are done for the small Haldane system with four unit cells, same as in Fig. 6(b) of the main text.

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
