# Peer review of "Feedback cooling of fermionic atoms in optical lattices"

_SciPost Physics, doi:SciPost Phys. 19, 073 (2025)_

## Round 2 · Referee Report · Anonymous (Referee 1) · 2025-5-15

Report

Zhao et al explore dissipative approaches to engineering topological band structures with ultracold fermions in optical lattices. This manuscript addresses a fundamental limitation of established methods based on adiabatic passage: a critical point has to be crossed in any topological phase transition at which there is no adiabatic timescale leading to imperfect quantum state engineering.

The explored approach, based on engineering dissipation using Markovian feedback control such that the desired final state is a dark state of the Lindbladian jump operator, builds on existing theoretical frameworks (e.g. the Wiseman-Milburn equation) and related proposals. Indeed, feedback control offers significant possibilities for cold-atom experiments given the feasibility of controlling millisecond dynamics using control actuators with bandwidths exceeding 1-10 kHz.

The submitted manuscript is well written, and the scientific content is well explained and employs sound methodologies. I particularly appreciate the authors' consideration of experimental limitations (namely, locality of measurement and feedback around few neighbouring sites) in designing approximate schemes that are more feasible for real-world implementation. While the manuscript does not address technical implementation of the proposed scheme, this does not detract from the contribution of the work which provides useful insight into the limits of the proposed approaches.

I believe this manuscript is appropriate for the target journal of SciPost Physics, provided the relatively minor comments and questions raised below can be addressed in a revised submission.

Specific questions and comments:

  1. gamma = 1e-4 is presumably chosen to satisfy the condition that the feedback Hamiltonian can be neglected. What is the error associated with the neglect of this term? Could it be, for example, estimated and compared to the infidelity predictions of the approximate method? If the error is on the order of 1%, for example, then comparison of infidelities at the 1e-3 level (as in Fig 3b) may not be appropriate.

  2. I would appreciate further elaboration on the control actuation, i.e. how the feedback operators can be implemented in practice. The authors mention, for example, that accelerating the optical lattice can be used to tune the complex tunneling amplitude between sites. Has this been demonstrated experimentally? What are the limits of this control, e.g. what limits how much support the control actuation on site A has from very distant sites on the lattice?

  3. For large systems, the authors adopt a mean-field approach (as opposed to direct numerical simulation of the master equation) to treat large systems in the Haldane model. The quality of the protocol is quantified by the deviation of the mean occupation numbers from the target ground state. I am curious as to how well the topological structure can be characterized by mode occupation alone, which does not capture coherences between different modes. Could the authors provide further clarification on the validity of characterizing topological insulator regimes using occupations alone? Furthermore, is there a relationship between the deviation D and the fidelity F that could be explored in simple cases where an exact analytical or numerical solution is tractable?

  4. At present, the validity of the mean-field approximation <n_k n_q> = <n_k><n_q> is not sufficiently addressed. Could the validity of the mean-field approach be validated by direct comparison to exact numerics for small system sizes? I expect that for the small system sizes the two will have a notable deviation, but in order to trust the mean-field approximation (i.e. neglecting non-trivial correlations) it would be useful to show that the two approaches converge for increasing system size.

Recommendation

Ask for minor revision

  • validity: -
  • significance: high
  • originality: high
  • clarity: top
  • formatting: excellent
  • grammar: perfect

Author:  Wenhua Zhao  on 2025-07-22  [id 5661]

(in reply to Report 1 on 2025-05-15)

We would like to express our appreciation for the referee's insightful comments, and for the constructive suggestions on the presentation of the results. We hope that the following response addresses the referee's comments persuasively:

  1. We understand the referee's concern. The feedback H_fb was not included in the simulations so far. We have now added a statement in the main text to clarify this as well as a quantitative analysis in Appendix A. To study quantitatively the impact of the measurement strength \gamma, we have added H_fb to the system Hamiltonian and computed fidelities for a small SSH model with three unit cells, considering different \gamma values and two different topological regimes: J1/J2=2 and J1/J2=0.5. The results are shown in Fig.1 attached, also included in the revised manuscript. The setup is the same as in Fig.2 of the main text for the approximate approach. By slowly changing \gamma from 0.0001 to 0.1, we observe that a smaller \gamma gives slightly higher fidelity, but in both cases the influence of \gamma in fidelity is on the order of 10^{-5}, thus justifying the omission of the H_fb term in the simulations.

  2. We thank the referee for their thoughtful question regarding the experimental feasibility of implementing the feedback operators, particularly the actuation part of the control. As noted, controlled acceleration of the optical lattice or periodic modulation (shaking) can introduce effective complex tunneling amplitudes. This technique has been used, for example, in the realization of the Haldane model (Jotzu et al., Nature 515, 237 (2014), Ref.7 in the main text), where complex NNN hopping was engineered using circular lattice shaking. On the other hand, the spatial support of the control (i.e., how many distant sites a feedback operation can coherently couple to) is limited by the ability to resolve and address those sites. For instance, site-resolved control is constrained by the diffraction limit and the spacing between lattice sites (typically on the order of hundreds of nanometers). Addressing long-range feedback operators (such as those involving delocalized Wannier functions) would require highly focused beams with programmable spatial light modulators or acousto-optic deflectors. For this reason, we introduced and emphasized the approximate scheme, where both the measurement and feedback operators are strictly localized. These operators involve only a few neighboring sites and are significantly more accessible to current experimental techniques. We have expanded the discussion in Section 3.3 to elaborate on these experimental aspects and cited relevant experimental works that have demonstrated the required ingredients.

  3. We fully agree that mode occupations alone do not capture the full topological structure of a many-body quantum state. However, our use of the occupation-based deviation D in the mean-field approach is not meant to characterize the topological phase itself, but rather to measure the proximity of the prepared steady state to the ground state, assuming the ground state is already known to be topological in character. This is a pragmatic proxy for cooling performance in large systems where direct fidelity calculation is not feasible. We agree that investigating the connection between the deviation measure D and fidelity provides valuable insights. To illustrate this relationship, we examine a representative case where the Wannier functions are localized at four sites within a four-unit-cell system, consistent with the configuration presented in Fig.6(b) of the main text. The attached Fig.2 shows that ( 1-D ) exhibits qualitatively similar trends to the fidelity. This parallel behavior justifies using D as a practical metric for evaluating our protocol's performance, particularly in large systems where fidelity calculations become computationally prohibitive. We have added a discussion and numerical illustration of this relationship in Sec.E in the appendix.

  4. We thank the referee for pointing out this important issue. Indeed, the validity of the mean-field approximation—namely the factorization < n_k n_q> \approx < n_k> < n_q> used in deriving the kinetic equations—is a key assumption for applying our approach to large systems. We agree that it is essential to assess the reliability of this approximation. In response to the referee's suggestion, we have performed exact diagonalization (ED) simulations for a small SSH system of 4 unit cells with J1/J2 = 2, computing < n_k n_q > (Fig.3(a)) and < n_k > < n_q > (Fig.3(b)). The difference between them (Fig.3(c)) is negligible, supporting the approximation < n_k n_q> \approx < n_k> < n_q>. To further validate the mean-field approach, we compared the deviation D obtained from ED and mean-field theory (Fig.4 attached). The close agreement between the two methods confirms the validity of the mean-field approximation. We have added a detailed discussion of these comparative analyses in Appendix E to address this comment more thoroughly.

Attachment:

Revision_Report1.pdf

---

## Round 2 · Referee Report · Anonymous (Referee 2) · 2025-6-21

Report

The manuscript by Zhao et al discusses the possibility to use feedback to produce an effective cooling procedure to prepare low energy states in topological insulator states for fermionic atoms in optical lattices.

The method in the paper follows closely three previous papers (Refs. 34-36) from the same last author. In all these papers, feedback is assumed as instantaneous and linear in the measured signal. This results in a time-local master equation, where the effect of the feedback process effectively becomes a standard dissipation channel. As such, this paper (and all the previous papers) are effectively determining the effects of a particular form of engineered dissipation on preparing a desired steady state. Feedback is considered as the physical origin of the engineered dissipation, but the actual starting equation (Eq. 7,8 of the current manuscript) is that of dissipation.

In the current manuscript, the goal is to achieve low energy states of a topological insulator with this process, and to realise this using feedback/dissipation that is local. The manuscript in fact presents an argument as to why this goal should fail: the proposed protocol relies on being able to define localised Wannier states, but in a topological state, Wannier states cannot be localised. From this perspective, the main result of the manuscript is the observation that despite this anticipated problem, a reasonable degree of cooling and low energy state preparation was indeed possible.

While the above results should be published in some form, I am not convinced there is a case for publication in SciPost Physics, as opposed to SciPost Physics Core. Of the possible criteria for SciPost Physics, I believe the most relevant would be "Open a new pathway in an existing or a new research direction, with clear potential for multi-pronged follow-up work". However, as noted above, the basic idea of feedback for engineering appropriate dissipative channels has already been presented by the last author in Refs. 34-36. In light of those prior works, I do not see a compelling case that this manuscript matches the criteria for SciPost Physics. As such I recommend it for publication in SciPost Physics Core.

Requested changes

  1. The manuscript work under the assumption that one can neglect the O(gamma) terms in Eq. 7 to produce a simplified model. This seems reasonable for the behaviour over a finite time range, but may lead to issues regarding the infinite time limit, as the long time and small gamma limits may not commute. Since real experiments are of finite duration, this does not necessarily change the validity of the calculations. The authors may wish to comment about this.

  2. In the general introduction to feedback control, the operator F is allowed to be arbitrary (Hermitian) operator. However around Eq. 13 a specific form of F is chosen. It is not explained why this particular form is chosen. Related to this, the manuscript does not sufficiently discuss how this form of F would be experimentally applied to the system. The authors should explain this point more clearly.

  3. After equation 18 there is a statement that optimisation can be done analytically for n=2. Is this meant to be a reference back to the n=2 form in Eq. 16,17? This should be clarified.

  4. For most of the models discussed, the authors show both the fidelity of the final state and the "minimum overlap". However Figure 3, for the topological pumping protocol, only shows the fidelity. For consistency with other figures, it would be useful to include minimum overlap vs theta in this figure.

  5. The use of the "c" operator in Eq. 37 is unnecessarily confusing. This notation appears only in this equation, and it would be significantly clearer to just write out the c=a, c=b cases explicitly. This would add one line to the equation.

  6. There is not much discussion of figure 5. Remarkably, while this is the "exact" protocol, the fidelity deviates from 1 notably for small phi. The discussion in the text implies that the exact approach ought to produce the exact answer. Can the authors clarify why this figure does not just show 1 throughout? This seems a surprising result that is just shown without comment.

  7. There are a few confusing aspects of figure 7. Firstly, the (green) curves for Delta=2J in the top left and bottom left panels seem to have opposite signs. Presumably there is a gauge freedom in the phase of the optimised wavefunction. As such, this may be a meaningless difference. Can one fix this gauge by setting the phase on site 11A to be positive real, to avoid such features? In addition: the different panels have different axis ranges (partly necessitated by this different sign); can these be consistent? I also did not understand why site 21B is placed as if it connects to 11B and 12A. I would have thought the logical order of sites would be 21B, 11A, 11B, 12A to match the figure.

  8. In appendix C, between Eq. 44 and Eq. 45 there are some confusing statements about Lindblad form. The manuscript makes the statement that it is necessary to remove off diagonal terms "to achieve an equation in Lindblad form". This is not correct, as discussed below.

As discussed in chapter 3 of the textbook by Breuer and Petruccione, the most general trace preserving Hermitian master equation will take the form:

\frac{d\rho}{dt} = -i [H_{\text{eff}}, \rho] + \sum_{a,b} K_{ab} ( X_a \rho X_b^\dagger - (1/2) [X_b^\dagger X_a, \rho]_+ )

where K_{ab} is the Lindblad-Kossakowski matrix, and X_a are a set of operators. I have written a,b to indicate the index required to fully define one of these operators. That means that in the authors notation, a is equivalent to the combination of k,q. and b to k', q'.

In writing this expression, there is a choice about which operators X_a one uses to write this. One may change from one set of operators to another by writing, e.g. X_a = S_{ab} Y_b, and this will then lead to an equivalent equation (i.e. one describing identical physics) written in terms of new operators Y, and a different matrix K, defined in terms of the old K and the transform matrices S. Because of this, one can always choose a transformation to diagonalise the Lindblad-Kossakowski matrix. However, for some matrices K, the resulting diagonal form may have some non-positive values. The meaningful question of whether a problem is of Lindblad form is about whether K is a positive matrix or not.

In some cases, where K is not a positive matrix, one may use a secularisation procedure to derive an effective K that is positive. This is the procedure the authors describe in the appendix, of removing terms from K that are off diagonal in the Hamiltonian eigenoperator basis. However, for the dissipation the authors are describing, their matrix K is in fact positive to start with. This is easy to see, since the dissipation starts in Lindblad form in Eq. 7, and this is just rewritten in terms of energy eigenoperators. In fact one may immediately see for the problem under consideration, that the K is a rank one matrix. That is, when diagonalised, only one of its eigenvalues is non-zero. That one non-zero eigenvalue corresponds to the single decay channel, C, in the original problem.

The summary of the above is that the manuscript takes an expression that is already of Lindblad form, and then performs secularisation on it with the claim that this is required to put it into Lindblad form. The secularisation process is not needed to be of Lindblad form, however it can be considered as reasonable approximation in order to make the kinetic theory derivation simpler. That is, the equations could remain if the justification was that this step was needed to simplify subsequent calculations, rather than implying that Eq. 44 is not of Lindblad form.

  1. The manuscript is selective in references to previous work on feedback control of many body/cold atom systems. In the current manuscript, the only references given (Refs 34-36) are all to the authors own work. However, in Ref. 34 there is a more appropriate set of references (Refs. 19-55 of Ref. 34) to the wide range of other work on measurement induced feedback in relevant systems (at least for papers up to 2022). The reference list on this topic should be broadened so as to not exclusively contain self citations.

Recommendation

Accept in alternative Journal (see Report)

  • validity: good
  • significance: ok
  • originality: good
  • clarity: good
  • formatting: excellent
  • grammar: excellent

Author:  Wenhua Zhao  on 2025-07-22  [id 5662]

(in reply to Report 2 on 2025-06-21)

We would like to thank the referee for taking the time to read our work very carefully and for providing valuable comments and suggestions. We address all points raised in detail in the following and hope that the referee will find our answers convincing.

Before that, we want to argue further in favour of the suitability of our manuscript for publication in SciPost Physics, as opposed to SciPost Physics Core. We agree with the referee that proposing the idea of using feedback mechanisms as engineered dissipation channels in cold atomic systems was the subject of previous articles by some of us, rather than being the key novelty of this work. However, the present work makes a different and substantial step forward, in the fact that: (i) We analyze in detail, for the first time, application to topological phases of matter and fermionic systems with the type of measurement and feedback envisioned; not only do we consider one-dimensional topological insulators, but we also explore two-dimensional systems, by analyzing a small-scale Haldane model with exact methods and larger scales with a mean-field approach. (ii) The interplay of topology, interactions (not present in the system, except for very weak interactions to break degeneracies, but arising from the Lindblad dissipators, which contain terms that are quartic in field operators) and controlled dissipation reveals interesting connections between these fields and requires the application of a broad range of technical methods; the connections include, for instance, the relation between Wannier functions and approximate cooling opportunities and the relation between global measurement and feedback and its effective impact in terms of interband quantum jumps. Method-wise, our work combines the construction of Wannier functions with the open-system description of continuous monitoring and feedback, and the use of many-body techniques such as our kinetic mean-field approach. We believe that these aspects can stimulate future work in directions which are different from our previous papers (which rather dealt with problems such as preparing non-topological states, controlling heat transport, simulating thermal baths), in particular regarding the connection between topological properties and engineered dissipation and the development of protocols to dissipatively prepare and stabilize topologically non-trivial systems. In our opinions, these arguments make a strong case for the suitability of our manuscript for SciPost Physics and for the indications we made upon submission on fulfilling journal expectations.

Reply to the Referee's requested changes:

  1. We thank the referee for raising this subtle and important point, which is closely related to a comment by Referee 1. In response, we have performed additional numerical simulations varying the measurement strength \gamma to directly evaluate the impact of the feedback Hamiltonian H_fb on the steady-state fidelity. This analysis is now included in the manuscript in Appendix A. To address the impact of the measurement strength \gamma, we have added H_fb to the system Hamiltonian and computed fidelities for a small SSH model with three unit cells, considering different values of \gamma and two different topological regimes: J1/J2=2 and J1/J2=0.5, as shown in Fig.1 attached. The setup is the same as in Fig.2 in the main text for the approximate approach. By slowly changing \gamma from 0.0001 to 0.1, we observe that a smaller \gamma gives slightly higher fidelity, but in both cases the influence of \gamma in fidelity is on the order of 10^{-5}, thus justifying the omission of the H_fb term in the simulations.

  2. We thank the referee for the helpful suggestion and the opportunity to clarify this point. In the revised manuscript, we now explain more explicitly how the feedback operator F is obtained after Eq.(12). Specifically, we first construct the jump operator C [Eq.(12)] to ensure that the target many-body ground state is a dark state of the dissipator, i.e., C|g> = 0. This guarantees that the target state becomes a steady state of the master equation. Within the Markovian feedback formalism, the collapse operator C is related to the measurement operator M and to the feedback operator F according to C = M - iF [Eq.(6)], where both M and F are required to be Hermitian. Combining this with its Hermitian conjugate C^\dagger = M + iF, we obtain the explicit expressions: M = 1/2(C + C^\dagger), F = i/2(C - C^\dagger), as given in Eqs.(13) and (14). This decomposition uniquely determines M and F from a given C. Regarding the experimental implementation of $F$: In the exact scheme, the feedback operator F is typically non-local in real space, reflecting the delocalized nature of the optimal jump operator C. Implementing such non-local operations directly is a significant experimental challenge with current technologies, as it would require coherent control over spatially separated sites. While one could envision engineered feedback using global fields shaped by spatial light modulators or programmable optical potentials, these approaches are still limited in resolution and scalability. Thus, the exact scheme serves primarily as a theoretical benchmark that demonstrates the best possible performance under idealized feedback. The more experimentally realistic approximate scheme is designed to preserve good performance while remaining implementable with current or near-term capabilities. In the approximate (local) scheme, the feedback operator F typically corresponds to local tunneling Hamiltonians acting within a small spatial region (e.g., a unit cell), with complex amplitudes. These can be implemented experimentally using techniques such as Floquet engineering. We now cite relevant experimental demonstrations (e.g., Jotzu et al., Nature 515, 237 (2014); Aidelsburger et al., Nat. Phys. 11, 162 (2015))). We have now included this discussion in the revised manuscript in Section 3.3. We thank the referee again for prompting this important clarification.

  3. We thank the referee for pointing out this ambiguity. The statement after Eq.(18) indeed refers back to the n=2 case introduced in Eqs.(16) and (17), where the jump operator C is constructed from mode operators defined in a two-site unit cell.

  4. We thank the referee for pointing this out. For consistency, we added minimal overlap in the main text, also shown here in Fig.2(b).

  5. We acknowledge that it is much clearer to write out the c=a, c=b cases explicitly. The corresponding equation (37) is updated in the main text.

  6. We attribute the discrepancy at smaller \Delta and larger \phi to the increasing near-degeneracies and shrinking level spacings (see Fig. 3 attached), where the eigenenergies are sorted in an ascending way and enumerated along the x-axis. Plots on the left side are the energy spectra in linear scale. Plots on the right side are in semilog scale and describe the energy difference between neighboring points of the left plots. As established in Section 3.1 of the main text, these near-degeneracies hinder the efficiency of the exact protocol. We have extended this discussion in the revised version.

  7. We thank the referee for pointing this out. We understand the confusion of the referee caused by the order of the sites and random gauge freedom. We have now fixed these aspects in Fig.7 in the main text as suggested. The amplitudes are now all positive and the order of the sites in the plot reads as 21B, 11A, 11B, 12A, as shown here in Fig.4.

  8. The reviewer correctly notes that the master equation in Eq.(44) is already in Lindblad form, as it was derived from Eq.(7), which preserves complete positivity. Our wording in the original manuscript (``to achieve an equation in Lindblad form'') was imprecise. The secular approximation (removing off-diagonal terms in the eigenbasis of H_eff was not required to ensure the Lindblad form but was introduced as a simplifying assumption for deriving the kinetic equations (Eq.(45)). We have revised the text to clarify this point.

  9. We appreciate the reviewer's observation regarding the references and fully agree that the manuscript should properly acknowledge prior work on feedback control. We have now expanded the reference list to include key contributions from other groups.

Attachment:

Revision_Report2.pdf

---

## Round 3 · Referee Report · Anonymous (Referee 2) · 2025-7-25

Report

The revised manuscript by Zhao et al. has addressed all the points regarding presentation that I presented in my previous report.

Regarding the question of SciPost Physics vs SciPost Physics Core, having read the report of the other referee and the authors response I think there potentially is a case that could justify this being published in SciPost Physics. The clearest case is that it presents a way to overcome the challenge of preparing a topological state in cold atom experiments. In that sense there is a case that this does meet the criteria of "Present a breakthrough on a previously-identified and long-standing research stumbling block". When considering this criteria, the question of how closely related this is to previous work by the same authors is less relevant: even if an idea had been extensively discussed, the realisation that such an idea could be applied in a different context to solve an new problem is clearly valuable.

I am not certain there is a strong case for this criteria, but I do recognise that this work does present a route, with strong evidence it should succeed, that overcomes a known "stumbling block" for such experiments. As such, in light of the other reviewers comments, I would not object to this being accepted in SciPost Physics on that basis

Recommendation

Publish (meets expectations and criteria for this Journal)

---

## Round 3 · Referee Report · Anonymous (Referee 1) · 2025-8-18

Report

The revised manuscript by Zhao et al. has addressed all the points I raised in my report. In particular, the benchmarking of approximate methods against exact numerical diagonalization strongly supports the validity of the reported results.

Recommendation

Publish (meets expectations and criteria for this Journal)

---

## Round 3 · Author Response

Dear Editor,

On behalf of all authors, we would like to thank you and the two referees for taking the time to evaluate our work. We are glad to see that the work is appreciated. We would also like to express our appreciation for their insightful comments, showcasing that they have very carefully read the manuscript, and for their constructive suggestions on the presentation of the results, which have significantly improved the readability of the manuscript.

We have read the reviewers' concerns, and replied to each report directly in SciPost. We have revised the manuscript accordingly, to implement as many of the comments as possible. We also took the opportunity to change the occasional word here and there to further improve clarity. All key changes in the manuscript are listed below. Furthermore, we are convinced that our manuscript is qualified to be published in SciPost Phys (rather than SciPost Physics Core) and the justification is given in the response to referee 2.

Sincerely,
Wenhua Zhao

---

## Round 3 · List of Changes

1. Section Introduction: we expanded the introduction on feedback control and included key contributions from other groups. We highlighted the idea of connecting topological properties and engineered dissipation protocols.
  2. Section 3.2, page 5: we explained more in detail how to obtain feedback operator and measurement operator.
  3. Section 3.3, page 6: we added the discussion about experimental implementation and the challenges.
  4. Section 4, page 7: we specialised the methods used to solve master equation (also added in the abstract) and addressed the omission of the feedback term.
  5. Fig 3: we updated the figure and added minimal overlap in 3(b).
  6. Eq. 37: we updated the Hamiltonian for clarity.
  7. Section 4.2.1, page 13: we added more discussion for Fig.5.
  8. Fig 7: we updated the figure by fixing the random gauge freedom and the order of the sites.
  9. Section 4.2.2, page 15: we mentioned the validity of mean-field approximation is studied in the Appendix.
  10. Acknowledgements: we updated the funding information.
  11. Appendix A: we added this section to clarify the impact of measurement strength on fidelity.
  12. Appendix D: we updated the discussion on eq.44 to address that it is already in Lindblad form.
  13. Appendix E: we added this section to justify the validity of mean-field approximation.
  14. References: we expanded the literature list.

---

## Editorial Decision

published